# A hybrid piezoelectric resonator-based DC-DC converter

Jae-Young Ko ⓘ , Wen-Chin B. Liu & Patrick P. Mercier ⓘ ✉

Piezoelectric resonators are becoming attractive alternatives to conventional magnetics in DC-DC converters due to their favorable scaling and manufacturing properties. However, the efficiency and current handling capabilities of baseline piezoelectric resonator-based DC-DC converters degrade at higher voltage conversion ratios due to charge utilization limitations imposed by topological operation. Here we present an Always-Multi-Path Embedded Flying Capacitor Piezoelectric Resonator-based DC-DC converter that uses flying capacitors to add both hybrid multi-path output power delivery features and to reduce the internal charge redistribution losses within the piezoelectric resonator. Specifically, the proposed integrated circuit modifies the optimal voltage conversion of the piezo network from 2:1 to 3:1 while adding a switched-capacitor output network that enables multi-path operation at all times, resulting in a net optimal voltage conversion ratio of 9:1 for the converter, with 4x improved output current. Fabricated in a 180 nm high-voltage CMOS process, the developed chip achieves a peak efficiency of 96.2% at a 48-to-4.8 V conversion ratio.

The rapid growth of artificial intelligence (AI), cloud computing, and the Internet of Things (IoT) has led to an exponential increase in power demand in data centers[1,2]. These advanced technologies require massive computational power, which in turn demand highly efficient power delivery systems. In response to this surge in demand, power distribution systems in data centers are transitioning from traditional 12 V bus voltages to 48 V bus voltages, enabling a 4x reduction in input current ($I_{IN}$) and, as a result, reducing conduction ($I_{IN}^2R$) losses in power lines by up to 16x. This, however, does not change the load voltage requirement (often 5 V or less), and thus either multiple DC-DC converters are required (e.g., 48 V to 12 V, and then 12 V to <5 V), which results in undesired cascaded losses, or, preferably, a single DC-DC converter that can directly support 48 V to <5 V conversion can be employed.

Design of DC-DC converters that can operate over such large voltage conversion ratios (VCRs) while maintaining high current density and efficiency is challenging. Most recent 48 V DC-DC converter literature utilizes magnetic devices (with optional flying capacitors) to achieve high conversion ratios and efficient power delivery[3–12]. While such designs have been successful in providing high efficiency and managing power conversion for various applications, they are beginning to approach fundamental electromagnetic performance limits: the sub-linear volume/frequency scaling properties of magnetic devices pose a design challenge in trading-off size, current handling capabilities, and efficiency[13,14]. Thus, traditional magnetics-based approaches face stiff challenges to further scale performance as load power requirements continue to rise.

In light of these limitations, researchers have recently begun to explore alternative energy storage elements that offer opportunities to further scale the efficiency and compactness of power converters. One promising alternative is based on piezoelectric devices[15–24]. Unlike inductors, which store energy in magnetic fields, piezoelectric resonators (PRs), for example, store and transfer energy through mechanical deformation and piezoelectric effects. PRs offer several advantages over traditional magnetic devices, including reduced volume due to their thin planar form factors and superior volume-frequency scalability, their ability to be easily batch fabricated, and their potential for direct integration onto silicon chips in future work. Importantly, the high coupling and quality factor ($k^2 \times Q$, where $k$ is the electromechanical coupling coefficient and $Q$ is the quality factor) of

Department of Electrical and Computer Engineering, University of California, San Diego, La Jolla, CA, USA. ✉e-mail: pmercier@ucsd.edu

PRs make them particularly attractive when attempting to design high-efficiency, high-performance power systems, especially in the context of next-generation power conversion technologies[13,15]. While piezoelectric transformer (PT)-based designs can achieve a high VCR[20], commercially available PTs[21] tend to be bulky and less efficient, whereas custom-designed PTs increase design and packaging complexity[22–24]. For these reasons, we focus our attention in this work to PRs instead of PTs.

Since piezoelectric resonators have only recently been adapted for use in DC-DC converters, considerable analysis, optimization, and design of materials, geometries, packaging, and circuit topologies are needed[16–19,25–34]. To start, the literature has broadly adopted the use of a Butterworth Van-Dyke (BVD) model[25,35,36], illustrated in Fig.1a, to model the primary mechanical resonance of a PR using equivalent electrical components ($L$, $C$, and $R$), along with a model of its piezoelectric static capacitance ($C_P$). Such a model can then be used to help assess design and optimization trade-offs. These equivalent values are also affected by the employed mounting method; further details on both are provided in the Supplementary Fig. 1.

As can be seen from the BVD model, PRs are resonant and block DC; as a result, PRs are not a direct replacement for inductors, and traditional circuit topologies used in inductive-based DC-DC converters are not appropriate—new circuit topologies are needed. Much

of the recent literature[26–34] has focused on variants of a four-switch seven-phase baseline topology, shown in Fig.1a, that operates in the inductive portion of the resonant region; this allows for soft switching, a technique preferred in high voltage systems that minimizes switching losses and enhances efficiency. However, despite the promising scaling properties of PRs, the efficiency of the baseline topology drops quickly at large VCRs[26] due to charge circulation losses, making its suitability for applications that require large VCRs questionable.

To make matters worse, commercially available PRs are not (yet!) optimized for power applications, and cannot operate robustly at the high current demands of modern data center applications. Specifically, the maximum current-carrying capability of a PR is determined by its physical properties[16], such as material, vibration mode, and geometrical design, as well as electrical excitation strategies, also known as operation principles. Typically, given a specific set of physical properties and operation principle, PRs tend to exhibit optimal performance near the resonant frequency, where the impedance begins to enter the inductive region(Supplementary Fig. 1). While the baseline topology enables such operation while elegantly allowing for soft switching, all of the converter's current is processed by the PR; thus, the maximum output current capability of the baseline topology can be no higher than the limited peak current the PR itself can handle. To compound matters, the larger the VCR, the shorter the power state of

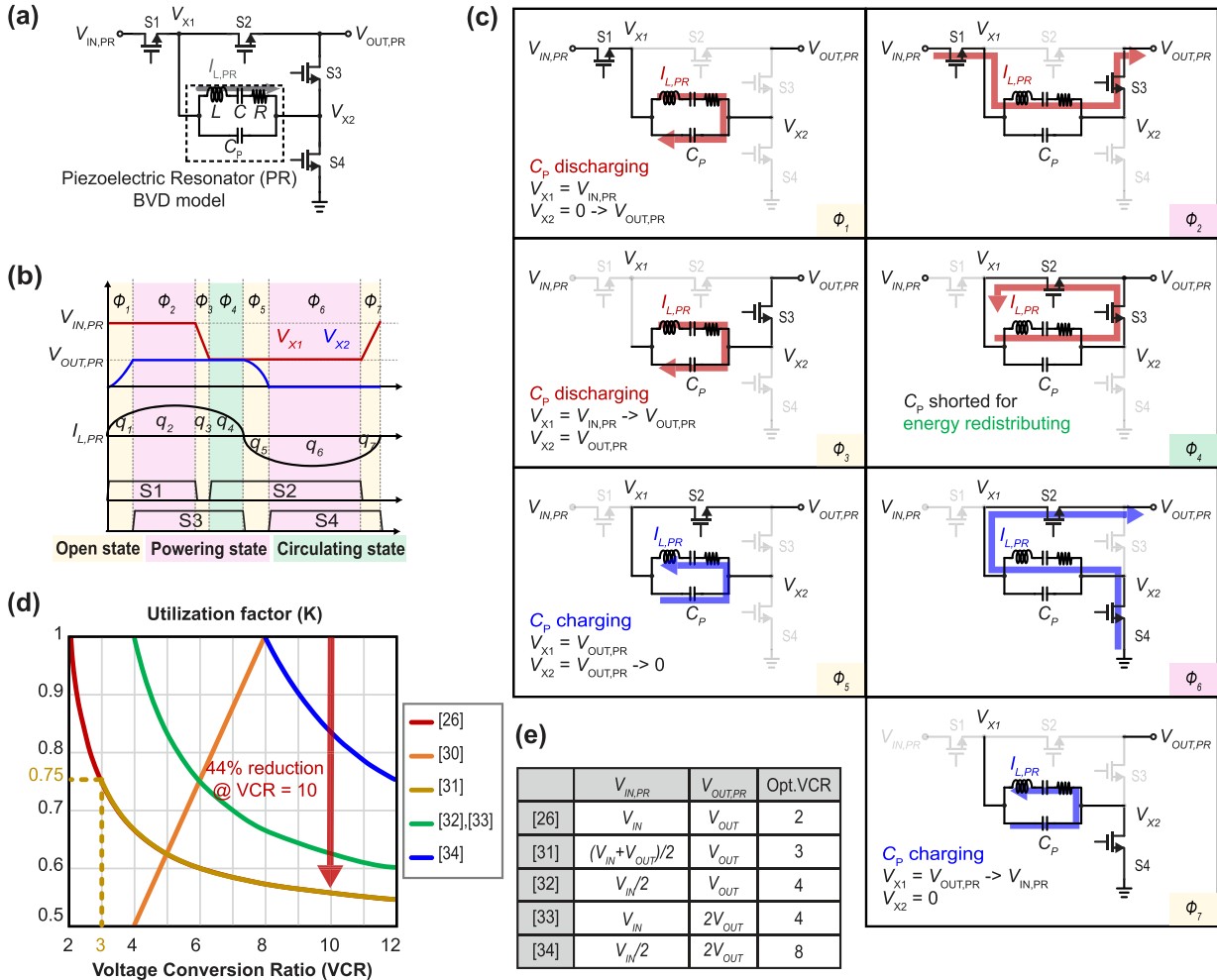

**Fig. 1 | Generalized previous works targeting lower VCR. a** Circuit schematic of a normalized PR-based converter. **b** Waveform of the voltage of both sides of the PR, the PR current, and power switch driving signals, following the switching sequence of $V_{IN,PR}$-$V_{OUT,PR}$, Zero, and $V_{OUT,PR}$. **c** Operation sequence over one period, following the switching sequence of $V_{IN,PR}$-$V_{OUT,PR}$, Zero, and $V_{OUT,PR}$. **d** Comparison graph of prior arts for the utilization factor ($K$), and $K$ degradation at higher VCRs (>10). **e** Comparison table with normalized expressions of $V_{IN,PR}$, $V_{OUT,PR}$ for topologies operating under the switching sequence of $V_{IN,PR}$-$V_{OUT,PR}$, Zero, and $V_{OUT,PR}$.

the baseline converter is, and thus the higher PR's peak current is, causing the PR to reach its maximum current limit more quickly than at lower VCRs. Because of these issues, PR converters have not yet exceeded the current densities achieved by state-of-the-art inductive-based converters used in data center applications. However, the high-$Q$ and promising scaling properties of PRs offer an attractive pathway to eventually get there. To help, new operation principles or topologies are needed to both reduce the current processing demands on the PR and reduce the circulating losses at larger VCRs.

In this article, we report a new circuit topology, implemented as an integrated circuit, that both reduces the amount of charge that needs to be circulated within the PR at larger voltage conversion ratios to improve efficiency, while also providing parallel power processing paths to augment the total amount of current the overall converter can process to improve current density at higher conversion ratios, ultimately helping get closer to the compact and efficient converter performance needs for next-generation data center applications.

## Results

### Prior-art PR-based DC-DC Converter Topologies

Prior work has enumerated multiple possible PR-based DC-DC converters utilizing varying combination of PR and switch positions and phases, ultimately yielding the topology shown in Fig. 1a as the option that delivers the best performance at the highest voltage conversion ratio (VCR = $V_{IN}/V_{OUT}$)[26]. In this configuration, the voltage across the PR terminals, $V_{CP}$, follows a switching sequence of $V_{IN,PR}$-$V_{OUT,PR}$, Zero, and $V_{OUT,PR}$. This topology, which uses four switches and a PR device, operates as shown in Fig. 1b, c over 7 phases in one of three PR states: (1) open, (2) powering, or (3) circulating. In the open state, which includes phases $\emptyset 1$, $\emptyset 3$, $\emptyset 5$, $\emptyset 7$, one side of the PR ($V_{X1}$ or $V_{X2}$) is open, and the parasitic capacitance $C_P$ is charged or discharged by the PR current, $I_{L,PR}$, allowing for soft switching. In the powering state, one side of the PR is connected to the output node, and another is connected to the input or ground, enabling power transfer to the output. In the circulating state, which only occurs during phase $\emptyset 4$, both sides of the PR are connected to each other, causing energy to circulate within the PR without transferring to the output, thereby allowing the converter to achieve continuous voltage conversion ratios.

This baseline converter achieves the highest efficiency at a 2:1 VCR, where the charge transfer utilization factor, $K$, which represents the ratio of charge delivered to the output to the total charge both delivered to the output and internally circulated, is unity. In other words, at a 2:1 VCR there is zero circulating charge, and the length of phase $\emptyset 4$ is zero. For VCRs larger than this, which are needed in modern data center applications as an example, the length of phase $\emptyset 4$ is increased in concert with a decrease in the length of phase $\emptyset 2$ (which deliver charge to the output), leading to more circulating charge (i.e., more losses) and a reduced $K$. As shown in Fig. 1d, the baseline topology's $K$ drops quickly at VCRs greater than 2:1. To make matters worse, the increased circulating charge at larger VCRs yields a larger sinusoidal $I_{L,PR}$, which results in higher PR vibration losses, significant efficiency degradation, and limited maximum output current capabilities. The latter point is especially important, as the total output current a PR-based DC-DC converter can achieve is ultimately limited by $I_{L,PR}$, and thus it is difficult to achieve high current density at large VCRs using a baseline topology.

To mitigate these issues, it is crucial to explore converter structures that can achieve high values of K even at VCRs higher than 2:1. Some prior art works published over the last few years have suggested including switched-capacitor (SC) structures to reduce the total amount of voltage and current that must be processed by the PR, ultimately improving $K$ at higher VCRs[30–34]. Generalized topologies of prior PR-based converter designs are provided in the Supplementary Fig. 2. Among these, a 2-stage cascaded topology[30] employed cascaded 4:1 SCs to achieve high $K$ values at higher VCRs (i.e., $K = 1$ at VCR = 8 as

shown in Fig.1d). By placing the PR stage at the input side and the SC at the output side, the PR can handle higher voltages and lower currents relative to the SC circuit, ultimately reducing $I_{L,PR}$ and thus improving both efficiency and output current handling capability. However, cascaded structures face some limitations, including independent operation of each stage, the need for multiple active and passive devices, and the presence of cascade losses, which typically result in lower overall efficiency. Moreover, this operating sequence cannot regulate to a VCR larger than 8.

Instead of simply cascading stages, merging the operation of the SC into the PR's natural operation, in a similar way to how hybrid inductive/capacitive converters operate[31–34], eliminates cascaded losses while providing the same benefits at large VCRs. These configurations modify the input and output voltages of the PR stage itself, $V_{IN,PR}$ and $V_{OUT,PR}$, using SC networks, but operate in the same sequence as the baseline structure introduced in Fig.1c. The optimal VCR where $K$ is maximal for various topologies is enumerated in Fig.1e.

While employing an SC network can help improve the optimal VCR, front/back/dual-side SC structures[32–34] are still constrained by the inherent optimal conversion ratio of 2:1 in the PR network itself, which these appended SC circuits do not modify. However, relying on operation near the PR's 2:1 ratio limits the ability to scale to larger VCRs, as the burden on the SC network increases and the soft-switching benefits of the PR at higher voltages are not fully utilized. While one merged SC approach[31] proposes a method for increasing the optimal conversion ratio of the PR network, this solution does not achieve $K = 1$ at the optimal operating point as shown in Fig. 1d, since both sides of the PR are connected to the output once every two cycles in phase 2, causing energy to circulate rather than being delivered to the output. This undesirable circulation not only reduces energy delivered to output but also induces additional power losses, making it difficult to achieve high efficiency with this topology.

### Embedded flying capacitors for modifying the PR network's optimal VCR

The challenge in modifying the optimal VCR of the PR network lies in maximizing $K$ (i.e., achieving $K = 1$ near the desired VCR) to ensure high efficiency and current capability. At the optimal operating point, baseline PR-based DC-DC converters[26,32–34] operate as shown in Fig. 2a (right), where, except for the open state that charges and discharges the PR's parasitic capacitor, $C_P$, all energy is transferred to the output, and the circulating charge ($q_4$) within the PR becomes zero. Baseline structures, which follow a switching sequence of $V_{IN,PR}$-$V_{OUT,PR}$, Zero, and $V_{OUT,PR}$, achieve $K = 1$ under these conditions, delivering the same amount of energy to the output during phases $\emptyset 2$ and $\emptyset 6$, while completely discharging $C_P$ during phase $\emptyset 4$. This implies that $V_{CP}$ are equal during both phases, and since $V_{IN,PR} - V_{OUT,PR} = V_{OUT,PR}$, the optimal VCR is 2. This result can be derived from the energy balance of the PR, expressed as $q_2(V_{IN,PR}$-$V_{OUT,PR})+q_6 V_{OUT,PR} = 0$, together with the charge balance of the capacitors $C$ and $C_P$, which gives $q_2 = $-$q_6$ at the $K = 1$ condition. Equations of charge balance and PR energy balance are provided in the Supplementary Note 1. Since the voltages in phases $\emptyset 2$ and $\emptyset 6$ are determined by the switching sequence, PR converters operating with this sequence[26,32–34] inherently have a fixed optimal VCR of 2 for the PR network.

In order to modify the optimal VCR of the PR without degrading $K$, here we propose an Embedded Flying Capacitor (EFC, $C_{EFC} \gg C_P$) structure as shown in the circuit of Fig. 2b, which can adjust $V_{CP}$ while ensuring a connection between the PR and the output during powering states. This EFC creates an additional DC node (other than $V_{IN,PR}$ and $V_{OUT,PR}$) via the DC voltage of the EFC, $V_{EFC}$, that can be used to connect the PR to. With this new DC node, the $V_{X1}$ node is no longer directly connected to $V_{OUT,PR}$, while its voltage rises to $V_{OUT,PR} + V_{EFC}$, and the PR can now operate with a switching sequence of $V_{IN,PR}$-$V_{OUT,PR}$, $V_{EFC}$, $V_{EFC} + V_{OUT,PR}$, allowing an optimal VCR, higher than the original, to be

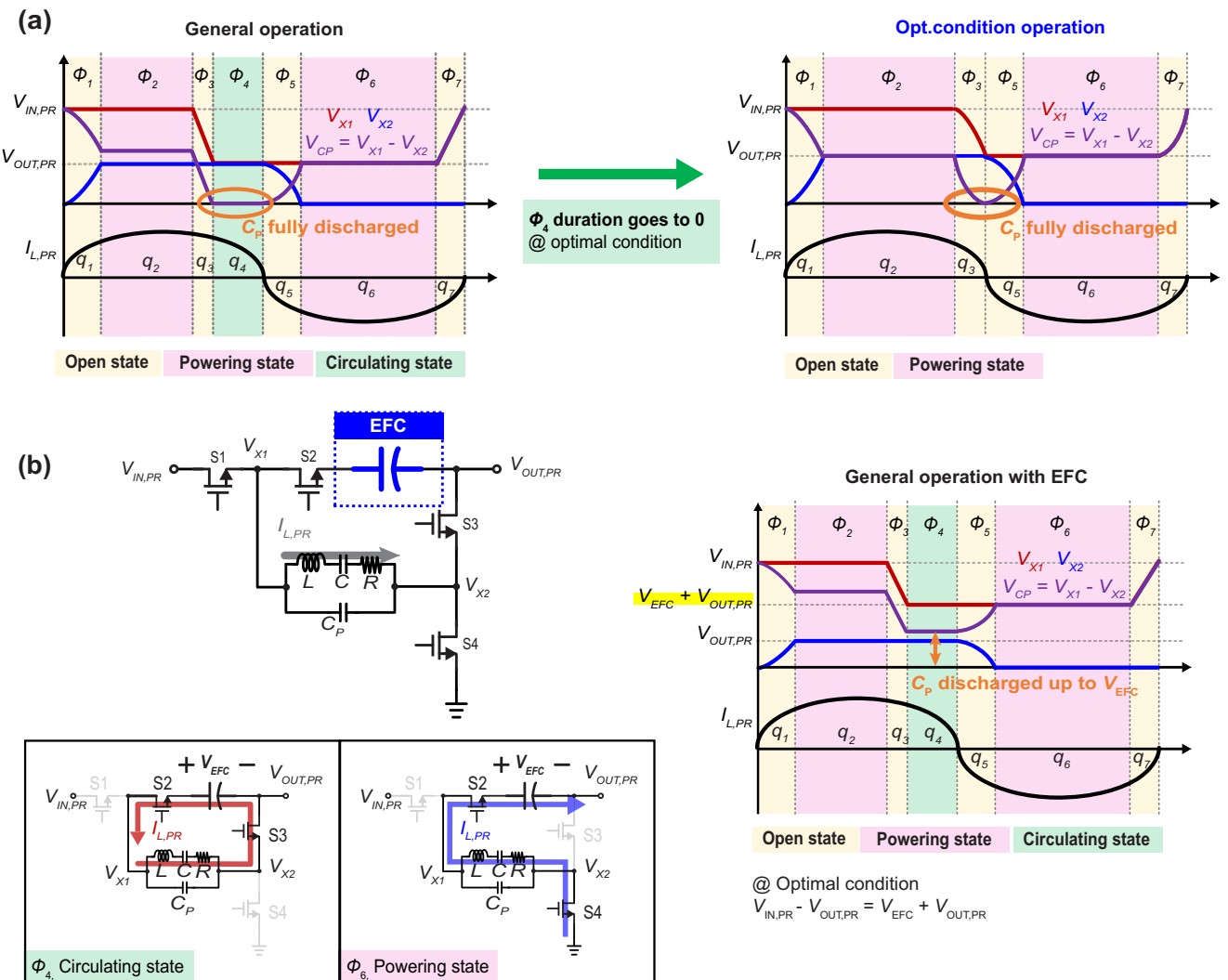

**Fig. 2 | Modifying optimal conversion ratio of the PR stage itself. a** Waveform of previous works under general condition and optimal condition. **b** Concept circuit applying an embedded flying capacitor (EFC), and operation under circulating state and power state (ø6) and voltage waveform of both sides of the PR, and the PR current of the EFC scheme.

derived based on the voltages in phases ø2 and ø6: $V_{IN,PR}$-$V_{OUT,PR} = V_{EFC} + V_{OUT,PR}$ (the optimal VCR is $2 + V_{EFC}/V_{OUT,PR}$). Ultimately, the EFC reduces charge circulation by making it unnecessary for the PR to be fully discharged, though it is partially discharged in phase ø4 and charged in phase ø6. Since phase ø6 is generally longer or the same compared to phase ø4, an additional path is required to charge balance the EFC, and furthermore this path requires that the powering state of the PR not be interrupted in order to avoid degrading $K$. Fortunately, setting the EFC voltage as a multiple of the output provides a simple solution that satisfies these needs, as it facilitates the creating of a discharging path toward the output and enables a self-balanced of EFC without additional control circuitry. Regarding the optimal VCR, if the DC voltage of the EFC is $nV_{OUT,PR}$, the optimal VCR can be changed according to the equation $V_{IN} - V_{OUT} = (n + 1)V_{OUT,PR}$, which results in an optimal VCR of $n + 2$.

This approach provides several advantages over traditional PR-based structures. By setting the voltage of the EFC to a multiple of the output voltage, the utilization factor can be maintained at 1 while allowing for a generalized increase of the optimal VCR of the PR network to be larger than 2. Furthermore, since the PR network can be positioned directly at the input side where it can favorably handle high voltages and low currents, this scheme does not impose limitations on current capability, helping improve current density. Additionally, by

preventing the full discharge of $C_P$, the energy used for charging and discharging $C_P$ is reduced by $2nC_PV_{OUT}$, which helps decrease the resonant current and improves efficiency.

This proposed configuration offers a promising way to enhance the performance of PR-based converters in modern power conversion systems, achieving higher efficiency and current capability, especially at higher VCR values. By adjusting the operational parameters and utilizing the EFC, it is possible to achieve optimized performance across a variable input and output conditions without compromising current handling capability or overall efficiency.

### Always-multi-path operation with backside SC network

While the proposed EFC technique helps to improve efficiency and current capability at high VCRs, the total output current handling capability of a PR-converter employing only the EFC technique is still ultimately limited by the finite current handling capabilities of the PR. As shown in Fig. 3a, the losses of DC-DC converters at heavy loads are dominated by conduction (i.e., $I^2R$) losses, and thus techniques that can help reduce either $I$ or $R$ can help further improve efficiency and/or achievable current density. For example, techniques that always distribute current across multiple paths in the pursuit of reducing such losses can be highly effective in boosting efficiency in traditional hybrid converters[37–49]. However, due to the resonant multi-phase

operation of PR-based converters, it is not straightforward to adapt such techniques to PR-based converters, and thus none have been described in the literature.

Interestingly, the PR-converter topology that utilizes a backside SC network[33,34], where the flying capacitor alternates between series and parallel connections with the PR network, as shown in Fig. 3b (left), technically provides a partial dual-path configuration. In such a configuration the PR network is placed in front of the SC network, and thus the PR is responsible for processing higher voltages and lower currents than if it were placed at the backside, therefore taking advantage of the favorable high-voltage/low-current properties of PRs. However, this structure only provides dual-path operation when the flying capacitor is connected in parallel with the PR network, which occurs for only half of the cycle. Even in the optimal condition, where the PR network delivers the same amount of energy during ø2 and ø6, the output current becomes unbalanced, as shown in Fig. 3c (left), and causes significant output voltage ripple. Note that baseline PR-based DC-DC converters exhibit a naturally discontinuous energy transfer characteristic due to their resonant nature, meaning that energy is transferred only during the powering states, which worsens output voltage ripple. To reduce output ripple, a paralleled PR-converter using interleaving[50] has been proposed. While this approach is effective in reducing output ripple and enhancing the overall system's current capability, it requires multiple PR devices, each of which involves complex mounting procedures, thereby limiting its ability to meaningfully increase current density.

Always Dual-Path (ADP) approaches, popularized in modern hybrid inductive/capacitive DC-DC converters[43–45], can provide a balanced energy transfer path under optimal conditions with two additional flying capacitors, significantly reducing both output current ripple and output voltage ripple. However, to date no prior-art PR-based converters have adapted ADP techniques due to the complexity of the multi-phase operation and the difficulty of merging an SC network in PR-based hybrid converters.

To overcome this challenge, here we propose an always-multi-path (AMP) structure that, by adding an additional flying capacitor that swaps positions with the one naturally present in the series/parallel backside PR-based topology, an AMP scheme that is compatible with PR converters and involves low overhead can be included. As shown in Fig. 3c, flying capacitor, $C_F$, in the partial dual path back-side approach is here split into two flying capacitors, $C_{FX}$ and $C_{FY}$, and switched in and out during phases ø2 and ø6. The energy storage and transfer of each flying capacitor helps reduce the burden on the PR network, which in turn lowers $I_{L,PR}$ by at least 2x, ultimately allowing for more power to be processed by the PR-based converter without approaching the inherent current handling limits of the PR itself. Moreover, this scheme provides a more balanced current path and results in smaller output current and voltage ripple compared to the partial dual path, as shown in Fig. 3c (right).

**Proof of concept and implementation**

To demonstrate the effectiveness of the proposed EFC and AMP schemes described above, we designed and fabricated an integrated circuit PR-based converter using the overall topology shown in Fig.4a. The converter consists of 13 power switches (S1-S13), 5 flying capacitors, and a PR device. Embedded flying capacitors, $C_{F1}$ and $C_{F2}$, adjust the optimal conversion ratio of the PR network, with $C_{F2}$ connected in parallel with $V_{M2}$ (equal to $V_{O,PR}$) during ø6, ensuring self-balancing: $C_{F1}$ balances by shunting $C_{F2}$ while maintaining $V_M$ as $2V_{O,PR}$. Thus, the converter achieves a 3:1 optimal PR network VCR with topological self-balancing that, when combined with flying capacitors $C_{F3}$ to $C_{F5}$ that form a merged backside SC network and thus provide an additional 3:1 step-down, result in a net optimal VCR of 9:1 in a PR-based converter. This high optimal conversion ratio coupled with the AMP approach

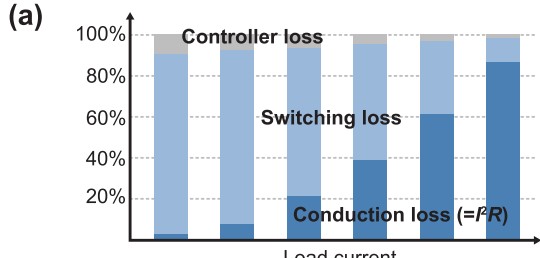

**(a)**

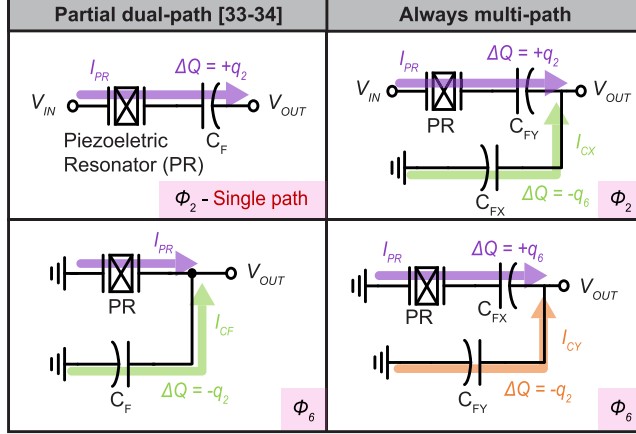

**(b)**

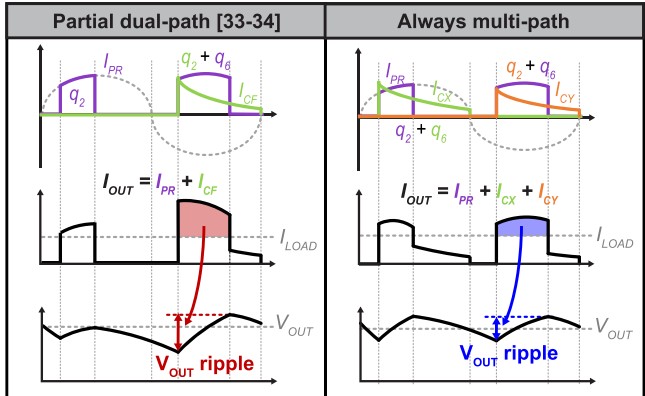

**(c)**

**Fig. 3 | Distributing current path. a** Loss breakdown of DC-DC converters according to load current. **b** Concept circuits of partial dual-path and always multi-path techniques. **c** Waveforms of output current and voltage ripple in both cases.

reduces the PR network's conversion burden while always transferring energy to the output in parallel with the PR, reducing ripples.

The converter operates over seven phases as shown in Fig. 4b with each phase in one of three states: (1) open, (2) powering, or (3) circulating. In ø1, which is an open PR state, $V_{X2}$ is floating, and since no current path is formed by the opened PR, $I_{L,PR}$ discharges the PR's junction capacitor, $C_P$, which increases $V_{X2}$. When $V_{X2}$ becomes equal to $V_{M2}$ ( = $3V_{OUT}$), switches S3,5,7,10 are turned on with zero-voltage switching (ZVS), whereas S11 and S12 turned on without ZVS. While S11 and S12 do not benefit from ZVS, their drain-source voltage, $V_{DS}$, is limited to $V_{OUT}$ ( < 5 V in our target), so hard switching is not a major concern in these switches. And energy transfer through the PR begins, starting ø2. ø2 is a powering PR state consisting of two power paths: one through the PR and one through $C_{F5}$. After ø2 delivers energy, S1 turns off and ø3 begins. During ø3, which is an open PR state, $I_{L,PR}$ discharges $C_P$ until $V_{X1}$ becomes $V_M$ ( = $6V_{OUT}$), then turns on S2 with

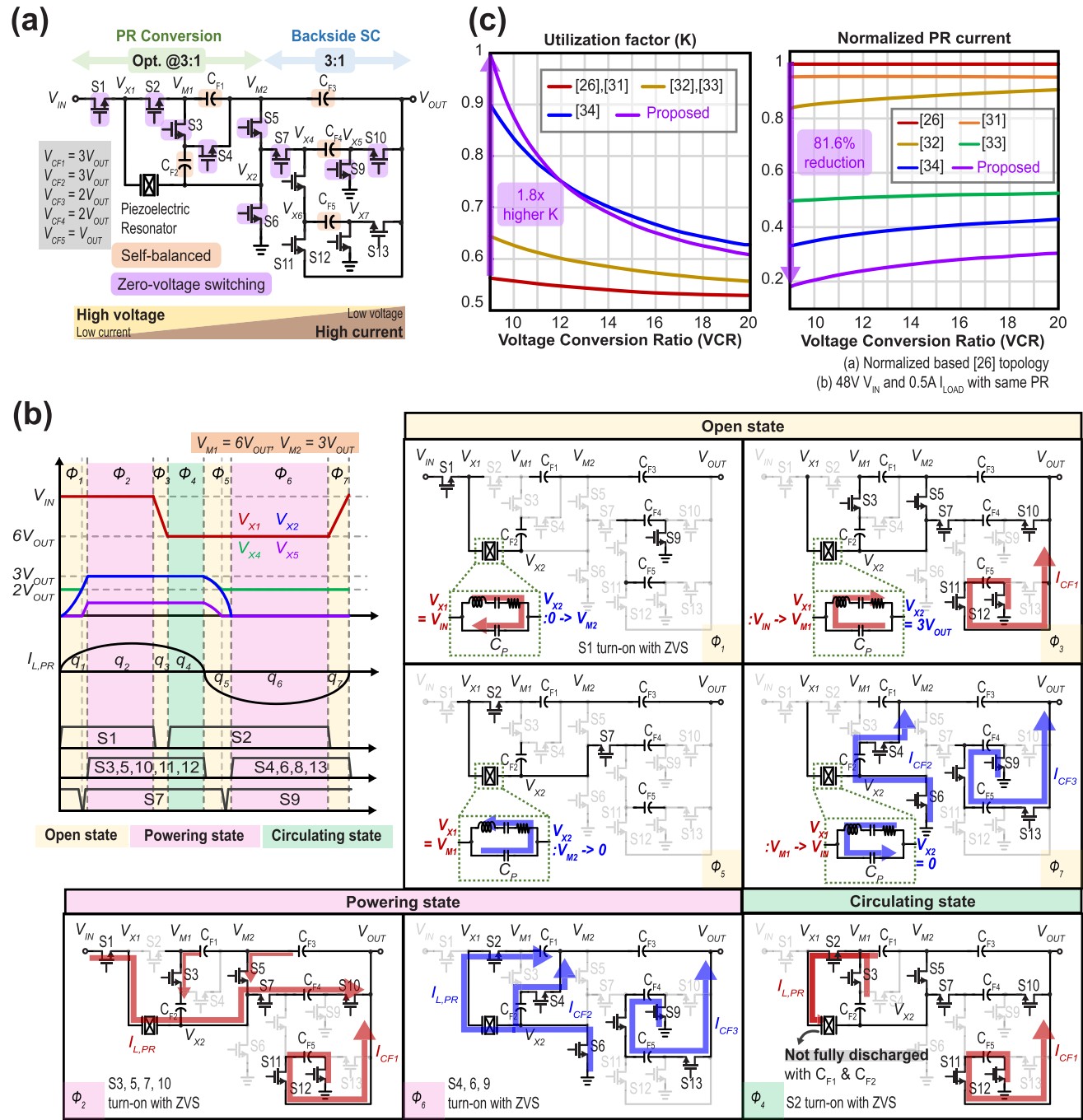

**Fig. 4 | Proposed converter applying EFC and AMP. a** Proposed power converter topology using EFC and AMP schemes. **b** Waveform of the AMP-EFC topology's switching nodes including voltages of both sides of the PR, the PR current, and power switch driving signals (left); operation sequence over one period (right). **c** Utilization factor (K) and normalized PR current amplitude in baseline PR-converter[26] compared to all prior art that can regulate VCRs >9.

ZVS, and initiates $\phi 4$. $\phi 4$ is a circulating PR state, where the PR current circulates and redistributes stored energy instead of going to the output. The timing of this state is used to set the output voltage to a desired target. Unlike conventional PR-based converters, which discharge $C_P$ completely, the proposed topology uses the EFC scheme with $C_{F1}$ and $C_{F2}$ to redistribute energy, thereby achieving an optimal PR conversion ratio of 3:1 without fully discharging $C_P$. Once the polarity of $I_{L,PR}$ changes $\phi 5$ begins, which is an open PR state, and S3,5,10,11,12 are turned off with zero-current switching (ZCS). The resulting negative $I_{L,PR}$ charges $C_P$ until $V_{X2}$ becomes 0. At this point, $\phi 6$ starts by turning on S4, 6 with ZVS and turning on S8, 13 without ZVS. Similar to S11 and S12, S8 and S13 has a limited $V_{DS}$ up to $V_{OUT}(< 5 V)$,

and thus, hard switching is not a major concern. $\phi 6$ is a powering PR state where energy is transferred by the PR, $C_{F2}$, and $C_{F4}$. Phase $\phi 7$, which is an open PR state, starts with S2 turning off, charging $C_P$ and increasing $V_{X1}$ to its initial state of $V_{IN}$. When the polarity of $I_{L,PR}$ changes, $\phi 1$ starts again. All flying capacitors are automatically balanced by repeating the series/parallel connection across cycles, eliminating the need for additional circuit for flying capacitor balance. Throughout these phases, the PR and/or flying capacitors ($I_{CF1}$, $I_{CF2}$, $I_{CF3}$) transfer energy together in a multi-path configuration, increasing efficiency and load capability and reducing ripple.

As shown in Fig. 4c, the utilization factor, $K$ at the optimal 9:1 conversion ratio is 1, which is a 1.8x higher than that of baseline PR

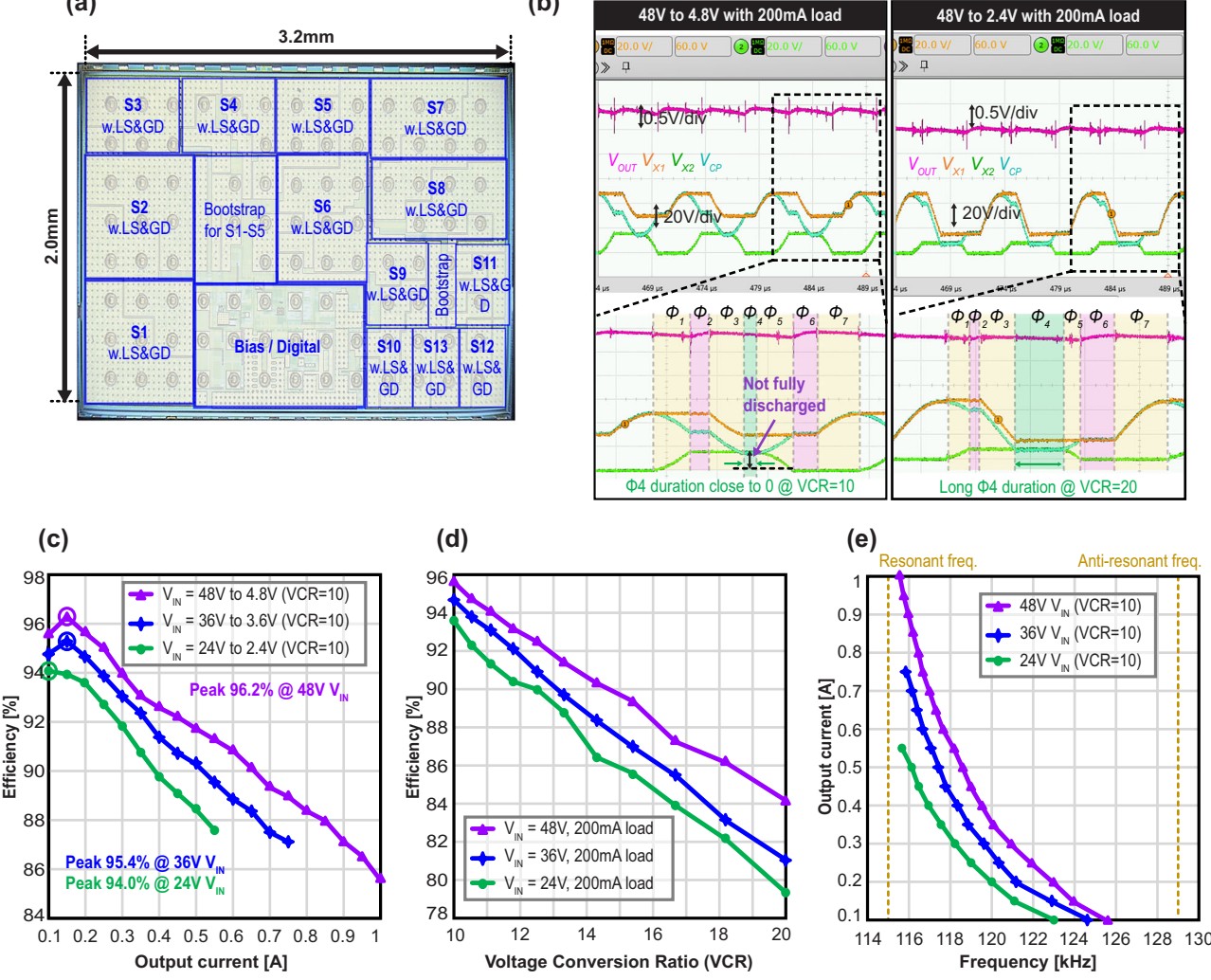

**Fig. 5 | Fabrication and measurements. a** Silicon die photo of the proposed converter. **b** Measured waveform of each side of the PR, it's differential voltage ($V_{CP}$), and output voltage under VCR = 10 and VCR = 20. **c** Efficiency curve versus load current with fixed VCR ( = 10). **d** Efficiency curve versus VCR with fixed load current (=200 mA). **e** Output current versus operation frequency, where the frequency operates in the inductive region of the PR.

converters. Moreover, the frontside positioning of the PR network and multi-path operation reduce the PR's current amplitude by 81.6% compared to the baseline PR converter under the same conditions as shown in Fig. 4c. The calculations for $K$ and PR's current amplitude can be derived from the amount of charge handled by PR in each phase and the energy distribution in PR and flying capacitors; these are provided in Supplementary Note 2 and Supplementary Fig. 3. Also, charge-based ripple analysis and simulation waveforms are provided in Supplementary Note 3 and Supplementary Fig. 4.

The proposed circuit's silicon implementation is shown in Fig. 5a, and details of the developed test printed circuit board are provided in the Supplementary Fig. 5. The silicon die occupies an area of 3.2 mm × 2.0 mm and includes 13 power switches, their driving circuits (level shifters, bootstrap circuits), and the bias/digital block. Detailed implementation of driving circuits is provided in the Supplementary Fig. 6. The chip was measured for various input voltages (i.e., $V_{IN}$ = 24 V, 36 V, 48 V), and different output voltages (10<VCRs<20)—where the output voltage was limited to <5 V due to the use of 5V-rated transistors (S9-S13) at the output stage of the fabricated IC—all within the PR's inductive operating frequency range of 115−129 kHz. The piezoelectric resonator employed in the experiment, made of PZT material, was selected with a high diameter-to-thickness ratio, which reduces characteristic impedance and enables higher current

operation at lower voltages[51], thus aligning with the design goals and supporting the performance of the proposed circuit.

The measured steady-state waveforms for $V_{IN}$ = 48 V and $I_{OUT}$ = 200 mA, with VCR values of 10 and 20, are shown in Fig. 5b. The individual phases are highlighted in Fig. 5b (bottom), where it can be seen that $V_{CP}$ does not discharge all the way to zero as in a conventional PR-based converter, indicating correct functionality of the EFC scheme. Also, under the condition where VCR = 10, it can be observed that the circulating state ø4 is shortened as the circuit approaches the optimal total VCR of 9 as shown in Fig. 5b (left), in good agreement with earlier analysis. Additionally, the balanced always-multi-path operation ensures that output voltage ripple during phases ø2 and ø6 shows similar behavior. Figure 5b (right) shows the converter operating under the same conditions except for an output voltage of 2.4 V (VCR = 20), where it can be observed that the circulating phase ø4 is lengthen as the circuit approaches the higher VCR, again in good agreement with earlier analysis.

The measured peak efficiency of the AMP-EFC PR-based DC-DC converter is 96.2% at a VCR of 10 for $V_{IN}$ = 48 V and $I_{OUT}$ = 150 mA, as shown in Fig. 5c. Figure 5c also shows peak efficiencies above 94% for other input voltages (24 V and 36 V) at the same VCR. Figure 5d shows the measured efficiency of the converter for varying VCRs and input voltages and a fixed 200 mA load current, demonstrating better than

**Table 1 | Comparison amongst state-of-the-art PR-based DC-DC converters using PZT**

| | TPE'21[26] | ECCE'23[30] | TPE'24[32] | APEC'24[33] | JSSC'24[34] | TPE'23[52] | This work |
|---|---|---|---|---|---|---|---|
| Technology | Discrete | Discrete | Discrete | Discrete | 180nm BCD | Discrete | 180nm BCD |
| # of stage | Single-stage | two-stage | Single-stage | Single-stage | Single-stage | Single-stage | Single-stage |
| Total opt. VCR | 2 | 8 | 4 | 4 | 8 | 2 | 9 |
| Opt. VCR of PR | 2 | 2 | 2 | 2 | 2 | 2 | 3 |
| VCR of SC | NA | 4 (backside) | 2 (frontside) | 2 (backside) | 2 (frontside) 2 (backside) | NA | 3 (backside) |
| Inductive region of PR [kHz] | 118 – 130 | NR [a] | 113–129 | 113–129 | 113–129 | 472–510 | 115–129 |
| PR size (Dia./Thick) [mm] | 19.8/0.8 | 29.9/2 | 20/0.2 | 20/0.2 | 20/0.2 | 4.75/0.67 | 20/0.25 |
| Power switch size [mm³] | 2x 1.8 (GaN) 2x 24 (Diode) | 4x 30 8x 2 | 7x 1.25 | 8x 1.25 | 1.83 | 4x 1.09 | 1.95 |
| Gate driver size [mm³] | 2x 3.2 | 4x 9.9 [b] | 7x 77.6 | 8x 77.6 | NA (on-chip) | 4x 3.2 | NA (on-chip) |
| Flying/bootstrap capacitors [uF] (size in inch) | NA | 5x 47 [c] (6075) | 1x 4.7 (0805) | 2x 10 (0603) | 3x 10 (0603) 9x 0.15 (0402) | NA | 4x 10 (1206) 1x 10 (0805) 9x 0.1 (0402) |
| Input [V] | 85–200 | 210–390 | 36–48 | 48–60 | 16–20 | 55–275 | 24–48 |
| Output [V] | 40 | 50 | 4–10 | 10–12 | 1.1–2.2 | 30–150 | 1.2–4.8 |
| Max. load [A] [d] | 0.25 | 1 | 0.3 | 0.5 | 0.25 | 0.0667 | 1 |
| Volumetric Current density [e] [A/cm³] | 0.80 (total vol.) 1.01 (only PR) | 0.32 (total vol.) 0.71 (only PR) | 0.47 (total vol.) 4.78 (only PR) | 0.72 (total vol.) 7.96 (only PR) | 3.52 (total vol.) 3.98 (only PR) | 2.31 (total vol.) 5.62 (only PR) | 8.43 (total vol.) 12.74(only PR) |
| Volumetric Power density [e] [W/cm³] | 28.71 (total vol.) 36.56 (only PR) | 15.99 (total vol.) 35.62 (only PR) | 4.71 (total vol.) 47.77 (only PR) | 7.20 (total vol.) 79.62 (only PR) | 7.75 (total vol.) 8.76 (only PR) | 69.38 (total vol.) 168.54 (only PR) | 40.48(total vol.) 61.15 (only PR) |
| Areal Current density [e] [A/cm²] | 0.06 (total area) 0.06 (only PR) | 0.08 (total area) 0.14 (only PR) | 0.05 (total area) 0.10 (only PR) | 0.08 (total area) 0.16 (only PR) | 0.08 (total area) 0.08 (only PR) | 0.17 (total area) 0.38 (only PR) | 0.29 (total area) 0.32 (only PR) |
| Peak efficiency | 98.0% [f] @85-to-40 | 96.2% @390-to-50 | 95.3% @48-to-10 | 95.8% @48-to-10 | 92.9% @20-to-2.2 | 98.0% [f] @55-to-30 | 96.2% @48-to-4.8 |
| Peak efficiency @VCR >9 | NR [a] | NR [a] | 89% @48-to-5 | NR [a] | 92.9% @20-to-2.2 | NR [a] | 96.2% @48-to-4.8 |

[a]Not reported.
[b]Not reported for gate drivers of SC stage.
[c]Size estimated from figure.
[d]Maximum output current at the voltage conditions of peak efficiency.
[e]Density calculated at the voltage conditions of peak efficiency, and total volume/area including PR device, power switches, gate drivers, and capacitors.
[f]Estimated from measured efficiency curve.

79% efficiency for VCRs as high as 20. The frequency variation for varying output current is monitored for all input ranges (24 V to 48 V) and a fixed VCR of 10 is shown in Fig. 5e. Figure 5e shows that the maximum current capability of the design is 1 A at a 48 V input voltage.

Table 1 compares the performance of the developed PZT-based PR converter to other state-of-the-art. Since our goal is to achieve a high VCR across the entire converter with high efficiency, we compare the topology from the enumeration paper[26] that demonstrated the highest VCR and operated with the same switching sequence of $V_{IN}$-$V_{OUT}$, Zero, and $V_{OUT}$, where $V_{IN} > 2V_{OUT}$, as used in other works. Given the variation in operating conditions across studies on PR-based converters, including differences in voltage levels, optimal VCRs, resonator dimensions, and operational frequencies, we confined our comparison to those using PZT devices. This approach clearly demonstrates the effectiveness of the proposed circuit topology without requiring separate materials-based optimizations. Furthermore, to ensure a fair comparison, volumetric current density, calculated from the maximum output current under the voltage conditions at which each work reports its peak efficiency and either total system volume or only the PR device volume, was selected as a key performance metric to normalize across different output voltages. This volume-based approach is particularly suitable for system comparisons, where the physical dimensions of the PR device directly impacts performance. Some designers may be interested in other performance metrics, however, and thus volumetric power density and areal current density—both using the total system volume/area and the volume/area of only the PR device—are also included for completeness. Notably, the proposed converter achieves both the highest total and PR-optimal VCR, along with the highest volumetric current density amongst all prior art using the same PR material: more than 10 times higher compared to discrete circuits implementations[26,32,33] that operate within a similar inductive region, over 3x higher than employing a smaller-volume PR device[52] with a higher switching frequency, and 240% higher compared to the only other published integrated circuit implementation[34], even though our design requires two additional capacitors. The two additional capacitors required cost less than 1 US cent each, making the cost impact negligible. Despite operating at a significantly higher volumetric current density, the proposed design operates at a 240% greater input voltage and a 13% improved optimal VCR, all while achieving a 3.3% improvement in efficiency compared to the only previously published integrated circuit used for this application. When considering the volume of the PR device only, the proposed design still achieves the highest volumetric current density compared to the art in Table 1. If designers are instead more interested in volumetric power density, the proposed work achieves higher density than all but [52] in Table 1, though it should be noted that [52] operates at lower VCRs and at output voltages >30 V, which is outside the scope of our intended application. At VCRs and output voltages needed for data

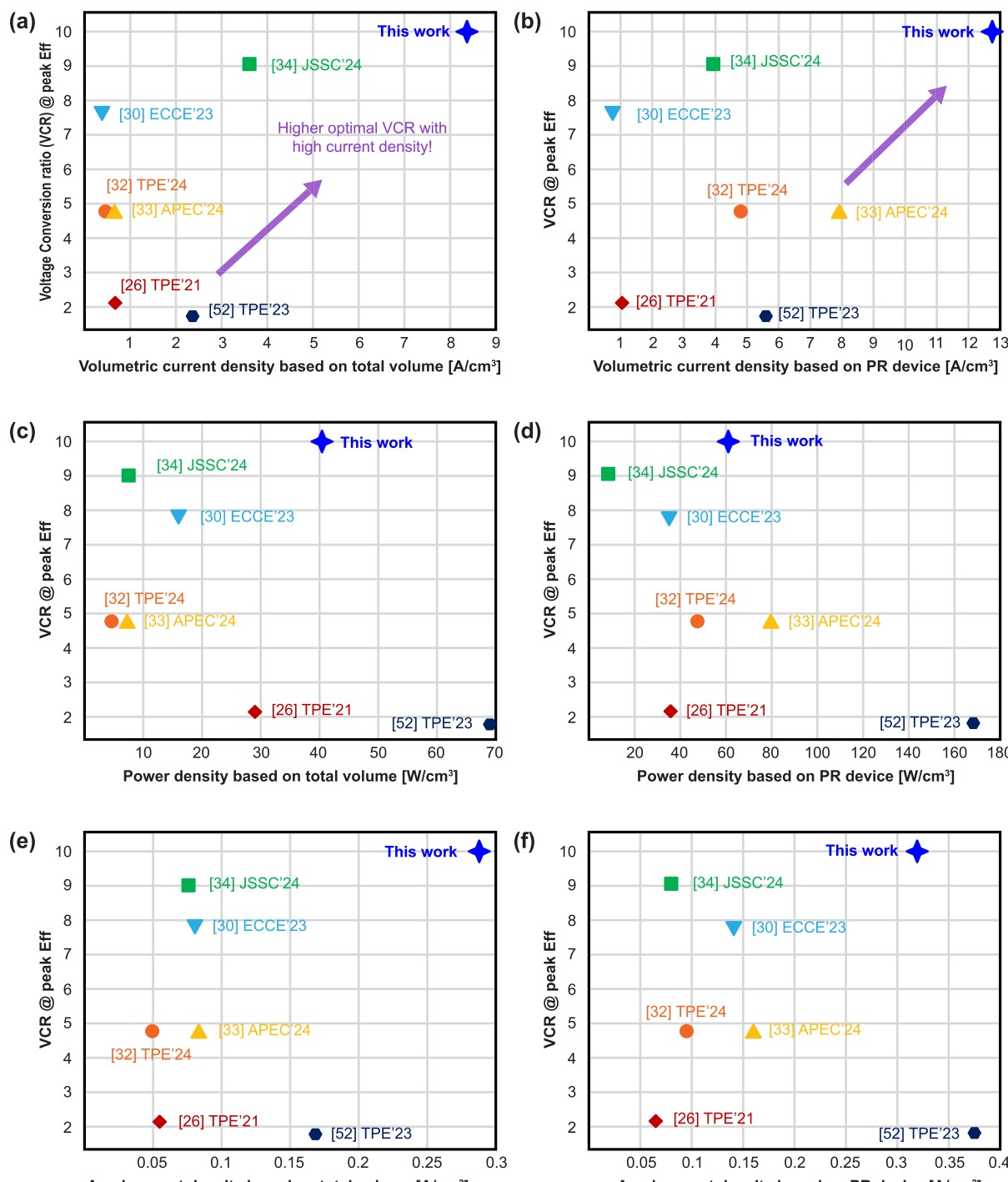

**Fig. 6 | Comparison of the VCR at peak efficiency and current/power density of the proposed design against state-of the-art PZT-based converters.**
**a** Volumetric current density based on total system volume. **b** Volumetric current density based on the volume of the PR device only. **c** Volumetric power density based on total system volume. **d** Volumetric power density based on the volume of the PR device only. **e** Areal current density based on total area. **f** Areal current density based on the area of the PR device only.

center applications, the proposed work offers a significant efficiency and volumetric current density improvement over any prior art as shown in Fig. 6a, though more scaling is still needed to meet modern data center demands. Figure 6b presents a comparison of volumetric current density based solely on the volume of the PR device, in order to highlight the impact of the circuit topology itself. Figure 6c, d shows the same data, but for volumetric power density, demonstrating that the proposed design doesn't achieve the highest power density, but

rather focuses on achieving excellent power density at the high VCRs and low output voltages needed in data center applications. Meanwhile, for designers who may be interested in applications that do not have z-height restrictions, Fig. 6e, f shows data for areal current density, where it can be seen that the proposed design achieves the highest areal current density based on the total system area, and slightly below [52] (though at a much higher VCR) when considering the area of the PR device only.

## Discussion

In this paper we have reported an AMP-EFC PR-based DC-DC converter. The proposed EFC scheme enables adjustment of the previously fixed optimal conversion ratio of a PR stage without limitations on current capability, $K$, or overall efficiency. Coupled with a proposed always-multi-path approach, which further improves current carrying capabilities, improves efficiency, and reduces output ripple, the proposed topology achieves state-of-the-art performance, helping pave a path towards the increased challenges found in high step-down conversion and high current density applications for data centers. Our integrated circuit implementation achieves robust high voltage operation at 2.4x higher input voltage than the only other reported integrated work, while eliminating the need for isolated drivers and external passive components, and simplifying PCB routing by internalizing all signal paths within the die compared to a discrete design. While the developed circuit was tested in an open loop configuration to enable rapid performance evaluation, which is the most common approach adopted in the literature (there are very few publications disclosing closed loop control of PR-based converters), future work will investigate closed-loop control. It should be noted, however, that since the proposed topology leverages a similar switching sequence to the baseline topology and other hybrid topologies built on the baseline topology, control techniques developed for such topologies can be leveraged here in future work. Additionally, further improvements in resonator materials, packaging, and device optimization, which ultimately limit the achievable current density, can be leveraged with this topology in future work to further improve performance. For example, the current density (based on only the volume of the PR device) achieved in [50] using lithium niobate is 25x higher than that of [26] and 4.6x higher than that of [52], despite employing the same baseline topology. Similarly, [52], which also uses PZT and the same baseline topology, reports a 5.6x higher density than [26] by employing a more optimized resonator device and operating at a higher frequency. These results serve as compelling examples of the significant potential that PR devices hold for future high-density power converters. In other words, the superior volume-frequency scaling properties of PRs make the future of PR-based DC-DC converters particularly attractive as more topologies begin to explore their capabilities.

## Methods

### Integrated chip and printed circuit board (PCB) fabrication

The chip shown in Fig. 5a was fabricated using a TSMC 180 nm BCD process, and is flip-chip bonded to a test printed circuit board (PCB). The chip includes not only the power stage, consisting of a power switch, bootstrap, and level shifter, but also a digital block and buffer to ensure correct timing of gate driving signal transmissions. Additionally, it contains a bias circuit that generates a reference current from an external voltage source. To operate the circuit at a 48 V input voltage, high voltage (HV) devices (<55 V) are required. The embedded flying capacitors reduce the voltage stress, $V_{O,PR}$ ( < 15 V), across S3-S6, ensuring a smaller PR conversion and allowing the bulky power switch of the 55 V device to be used only for S1 and S2. The PCB used for measurements was fabricated by Candor Circuit Boards, who provided highly precise 110 μm trace width and 50 μm spacing. The PCB is 100 mm by 100 mm in size and includes the PR mounting (Supplementary Fig. 5), all active areas, headers for input voltage (48 V $V_{IN}$, 5 V $V_{DD}$), three enable switches, and SMA connectors for verifying the impedance of the PR mounting.

### Piezoelectric resonator device and characteristics

The piezoelectric device used in the measurement was manufactured from APC International. The piezoelectric resonator is a disc-shaped product with a size of 20/0.25 mm (diameter/thickness) and made from 841 material. The device was mounted according to the method

provided in the Supplementary Fig. 1b, and the S21 parameter was measured using the E5080A (Keysight). The information obtained from the PR device using this method is presented in the Supplementary Fig. 1c, confirming that the inductive operation region of the PR is between 115 and 129 kHz.

### Passive components characteristics

This structure, which includes numerous switches for efficient conversion, requires several passive components. First, the 5 flying capacitors and 1 output capacitor included in the power topology are all designed with a target value of 10μF. Depending on the DC voltage rating, $C_{F1}$-$C_{F4}$ use 1206 capacitors (C3216X5R1H106K160AB), while $C_{F5}$ and $C_{OUT}$ use 0805 capacitors (GRM21BR61H106KE43K). Additionally, for driving power rail generation, 9x100nF 0402 capacitors (C1005X5R1A104K050BA) were used. The volume of these capacitors, the integrated chip, and the PR were calculated to obtain the power density.

### Measurements setup

The AMP-EFC converter proposed in this article operates under open-loop control to facilitate efficient parameter sweeping, implemented by a C2000 series DSP (F28379D). The input timing data for the DSP is determined by the following steps: first, the BVD parameter for the given PR and mounting is obtained using the E5080A (Keysight). Once the parameter is extracted, the PR resonant current amplitude, $I_{L,PK}$, is calculated using the provided equation in the Supplementary Note 2. Finally, the timing is calculated based on the previously obtained BVD modeling value and $I_{L,PK}$, and following the algorithm described below. Figure 4b illustrates the turn-on/off timing for each signal (S1-S13). Signal S1 is activated at the beginning of the cycle and deactivated based on voltage regulation. Signal S2 engages with ZVS turn-on at the beginning of ø4, when $V_{X1}$ falls to $V_M$ (equivalent to $6V_{OUT}$), and disengages according to the duration of ø7, ensuring that $V_{X1}$ rises to $V_{IN}$ by the end of ø7. Signals (S3, S5, S10, S11, S12) and (S4, S6, S8, S13) operate as out-of-phase pairs within a cycle. These signals initiate with ZVS when $V_{X2}$ rises to $3V_{OUT}$ (equivalent to $V_{O,PR}$) or falls to 0, and terminate at the half cycle or the end of the cycle. Signals (S7, S9) turn on slightly earlier than S3 or S4 with ZVS when $V_{X2}$ rises across $V_{X4}$ or $V_{X5}$ falls to 0, and turn off, ensuring dead-time for S7 and S9. The calculation of on/off timing can be derived with measured $C_P$, and calculated $I_{L,PR}$, and this computation is provided in Supplementary Note 4 and Supplementary Fig. 7.

Although the AMP-EFC converter includes an EFC, it still follows the same phase sequence as the baseline PR-based DC-DC converter[26]. As a result, it remains fully compatible with standard PR control schemes, enabling straightforward implementation of closed-loop operation.

## Data availability

The data that support the findings of this study are presented in the manuscript and Supplementary Information, or available from the corresponding author upon reasonable request. Source data containing the raw numbers are provided with this paper. Source data are provided with this paper.

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

## Acknowledgements

This project was supported in part by the Power Management Center (PMIC) an NSF I/UCRC, award number 2052809.

## Author contributions

J.-Y. Ko and P.P. Mercier conceived the overall concept, J.-Y. Ko designed the chip and circuit board, and completed testing. W.-C.-B. Liu designed and evaluated the PR mounting strategy. J.-Y. Ko and P.P. Mercier wrote the paper. All authors edited the paper.

## Competing interests

The authors declare no competing interests.
