## [Transparent Peer Review file · Nature Communications]

A Hybrid Piezoelectric Resonator-based DC-DC Converter

Corresponding Author: Professor Patrick Mercier

Version 0:

Reviewer comments:

Reviewer #1

(Remarks to the Author)

This reviewer sees the most noteworthy contribution of this paper as being the use of “embedded flying capacitors” in piezoelectric resonator-based converters to enable higher voltage conversion ratios with high efficiency. Other aspects of this work (hybridizing piezoelectric-resonator-based converters with switched capacitor networks, implementing power converter circuits and control on an IC, etc.) have been previously investigated. This reviewer has the following concerns:

1. There are some inaccuracies and incorrect terminology in the introductory descriptions of piezoelectric components. Line 50 - the provided definitions for C_p and C are incorrect. C_p is the static capacitance (i.e., the portion of the parallel plate capacitance that does not participate in mechanical resonance), and C is a modeled mechanical (sometimes referred to as motional) capacitance. Both combine together to form the parallel-plate capacitance at dc. Line 63 - the term "saturation current" is not applicable to PRs, so it is unclear what the authors mean by this in a physical sense.
2. The authors claim that some things have not been explored before that actually have. Line 190 - the authors report that "no work has reported on the output voltage ripple of PR-based DC-DC converters," but this was explored via interleaving in "A Spurious-Free Piezoelectric Resonator Based 3.2 kW DC–DC Converter for EV On-Board Chargers" by Stolt et al.
3. The comparison provided in Fig. 6 is a bit tenuous, or perhaps the authors' parameters for the comparison are not clear. While it's true that [24] does not have a high VCR or current density, its peak efficiency (99% in Figs. 24, 27, and 28) is misrepresented. For the current density comparison, there have been more recent designs of the baseline topology with greater density such as "A Piezoelectric-Resonator-Based DC-DC Converter Demonstrating 1 kW/cm³ Resonator Power Density" by Boles et al.
4. Some “for the first time” claims are a bit oversold. For example, the net optimal VCR of 9:1 is only a small improvement over previous designs with optimal VCRs of 8:1, and the methods used in those designs are easily extendable to higher conversion ratios.
5. Even though it is very thin, the piezoelectric resonator seems to take up a lot of surface area, so it is unclear how practical this approach actually is without significant development of piezoelectric resonator itself. The authors do not seem to have innovated on that aspect of the proposed system, yet it seems like the largest bottleneck.

Reviewer #2

(Remarks to the Author)

Authors propose a modified version of the baseline piezoelectric-resonator-based DC-DC converter which makes it hybrid and efficient thanks to the introduction of the always-multi-paths with backside SC network.

The novelty degree is sufficient for the publishing of the work. The paper is well-written and structured and technically sound. The noteworthy results relies on the novel approach using the embedded flying capacitor (EFC), which allows to shift the VCR to achieve the maximum charge transfer utilizing factor k , and the use of the always-multi-path for balanced energy transfer in the circuit. The experimental results are in-line with the carried out analysis and comparison with the state of the art is provided.

My only one comment is about the comparison with the state of the art which require, in my opinion, more works. Further, the

number of external devices should be reported in the comparison. How did the Authors calculate the current density for discrete component solutions? The comparison by using this metric results too far lookint at the compared works. In conclusion, comparison with prior art, which is very important for the proposal of the work, should be better stated.

Reviewer #3

(Remarks to the Author)

This manuscript demonstrates a groundbreaking integrated circuit for piezoelectric resonator (PR)-based DC-DC conversion operating at a 48V input—remarkably achieving state-of-the-art efficiency and current/power density. Such high-voltage capability is a pioneering result that will likely open exciting new directions in next-generation data center power conversion. The paper is well structured, explaining both the Embedded Flying Capacitor (EFC) method and the Always Multi-Path (AMP) topology in a clear, logical manner. Given its strong experimental validation and its potential to advance the field significantly, I highly recommend this manuscript for publication in Nature Communications. Although the current draft is well-prepared, I would like to highlight a few minor comments below. I hope you consider these as constructive suggestions to further improve the paper and make your valuable contribution even more impactful.

1. If there is sufficient room in the manuscript, consider adding a concise figure or table summarizing representative PR-based converter configurations. For instance, show how previous studies have used frontside switched-capacitor (SC), backside SC, or cascaded stages, then highlight each configuration's inherent optimal voltage conversion ratio (VCR) and limitations. Such a quick visual comparison would help readers grasp precisely how your EFC + AMP approach diverges from and improves upon existing methods. For example,
[28] Two-stage: PR-based converter (2:1) + SC converter (2:1)
[32] Single-stage: Frontside SC (2:1) + PR conversion (2:1) + Backside SC (2:1)
[This work] Single-stage: EFC-based PR conversion (3:1) + Backside SC (3:1)
2. In the early theoretical sections, the manuscript states that the EFC voltage is set to a multiple of the output voltage to shift the PR's inherent 2:1 ratio to a higher value (e.g., 3:1). Clarifying why this choice (as opposed to another reference voltage) is crucial to realizing the $K=1$ condition would further strengthen the reader's understanding of the design trade-offs.
3. While the innovation emphasizes large step-down (from 48V to a few volts), it would be helpful to briefly mention the highest feasible output voltage (e.g., under a 48V input) that still maintains the proposed benefits—such as soft-switching, high efficiency, and $K=1$.
4. You note that AMP improves current density by distributing current among multiple paths and reducing PR current amplitude. Another advantage likely includes reduced output ripple via more continuous energy delivery. Please consider providing a comparison—either numeric or waveform-based—showing how AMP specifically lowers output ripple compared to standard single- or partial-path approaches.
5. The text states that all switches are zero-voltage switched, but lines 226 and 238 raise questions about whether S8, S11, S12, and S13 always meet ZVS criteria. Including an explicit on/off timing diagram or notation in Figure 4(b) confirming which switches are ZVS-enabled—and at which phase transitions—would enhance the credibility of the soft-switching claim.
6. One key advantage noted is the minimized normalized PR current. As lower resonator current could mitigate vibration losses, reduce temperature rise, and improve reliability, please elaborate on how these improvements translate into higher output power capability or longevity. This will reinforce the importance of suppressing PR current amplitude. I guess that lowering the PR current amplitude correlates with higher maximum output current. Briefly elucidating the mechanism (e.g., PR saturation or Q-factor constraints) that dictates this upper bound on output current would strengthen the understanding of the design trade-offs.
7. Verify the labels on the horizontal and vertical axes. The intended label seems to be “VCR = 10” (not 0.1) for the x-axis, and “Output Current [A]” (not [mA]) for the y-axis. Correcting these minor issues will ensure clarity in your otherwise excellent measurement results.
8. To strengthen the work's real-world relevance—especially for data centers—compare your results against at least one state-of-the-art inductor-based solution operating in the same high-voltage, high-step-down domain. Demonstrating that a PR-based converter can rival or outperform existing inductive legacy solutions (in terms of efficiency or volumetric density) will highlight the practical significance of this novel PR-based topology.
9. In Figure 6, please clarify how you compute the converter volume—e.g., whether it includes the PR device's thickness, external capacitors, IC packaging, etc. Explaining these details will help readers fairly compare your density metrics with those in other works.
10. The paper employs an external DSP to run the converter under open-loop control. Since this circuit appears fully capable of closed-loop regulation, it would be helpful to clarify why you chose an open-loop scheme for validation. Was the DSP approach simply more convenient for rapid prototyping or parameter sweeping? Addressing this question will assure readers that closed-loop control is indeed feasible and that open-loop operation was selected primarily to streamline experimental demonstration.

Reviewer #4

(Remarks to the Author)

This manuscript presents a new circuit topology for a piezoelectric-resonator-based dc-dc converter. This circuit topology is derived by introducing an embedded flying capacitor (EFC) within the structure of a well-established circuit topology for piezoelectric-resonator-based dc-dc converters. The concept of introducing the EFC is similar, though not identical, to the concept of introducing a series capacitor in a buck converter to form the series capacitor buck converter. The EFC enables the proposed dc-dc converter topology to reduce the amount of charge that needs to be circulated within the piezoelectric-resonator at larger voltage conversion ratios. This is the main innovation presented in this manuscript. The authors also present an implementation of the proposed converter and its measured performance. Below are specific comments and questions for the authors:

1. Given that the introduction of the EFC is the main innovation of this manuscript, the authors should explain more clearly, including intuitively, how the presence of the EFC achieves a reduction in charge circulation at larger voltage conversion ratios. The current explanation related to the optimal voltage conversion ratio (VCR) for the conventional piezoelectric-resonator (PR) converter and the proposed PR converter provided in the first two paragraphs of the section titled "Embedded flying capacitors for modifying the PR network's optimal VCR" is not clear.
2. The authors should also discuss any limitations or constraints imposed on the design or operation of the PR converter by the introduction of the EFC.
3. The authors should also describe how the required value of the voltage across the embedded flying capacitor is determined for a given input voltage, output voltage and any other parameters of the proposed PR converter.
4. The authors should describe which components are included in the calculation of current density, and comparison with the state-of-the-art, in Fig. 6. For example, is the area of the piezoelectric resonator and the other passive components included in this calculation and comparison.
5. The authors should describe the thermal management strategy utilized for the converter during its testing. The authors should also comment on the factors limiting the testing of the converter at power, or current, levels higher than those at which the converter was tested.
6. The authors note that one strategy currently being employed or considered for 48 V to point-of-load conversion in applications such as data centers is the use of hybrid magnetic switched capacitor converters. The authors should also compare the performance (power density and efficiency) of their proposed hybrid piezoelectric-resonator converter with the state-of-the-art hybrid magnetic switched capacitor converters.
7. Given that the main application the proposed converter is targeting is data centers, and the power (and output current) requirement of data center point-of-load converters is increasing (and in some cases already around 1000 A), the authors should discuss the scalability of the proposed converter to such current levels, and how the scaling laws of hybrid piezoelectric-resonator converters compare with the scaling laws of hybrid magnetic switched capacitor converters.

Reviewer #5

(Remarks to the Author)

Version 1:

Reviewer comments:

Reviewer #1

(Remarks to the Author)

The revised version of the manuscript is improved but still has significant issues.

1. This work claims to be a solution for data center environments, but 48-5V (or 48-1V) data center power delivery tends to require kW (and often kA) power levels. The solution presented in this work is 2-3 orders of magnitude lower than this, so it is difficult to justify as a solution for this application.
2. There are multiple unclear descriptions and confusing terminology choices in the introduction. A) The authors mention that power distribution systems are transitioning from 12V to 48V, and that this allows for the delivery of higher current in a more compact form factor. However, transitioning to higher voltage should reduce current for the same power level, so this sequence of sentences should be clarified. B) It is unclear what the authors mean by "iso-performance" when referring to magnetic components. C) The term "energy conservation" should probably be "energy storage" when referring to piezoelectric components.
3. There are multiple inaccuracies in the introduction that are concerning. A) There is no such thing as a physical turns ratio

in piezoelectric transformers; as written, this sentence risks confusing readers. There is an ideal transformer in the model for a piezoelectric transformer, and there are many factors (material properties, geometric dimensions, etc.) that determine its transformation ratio. B) It is not clear what the authors mean by "beyond which they do not operate correctly" when referring to the maximum current of a piezoelectric component. Piezoelectric components do not have a maximum current rating; the relevant physical limits are mechanical stress/strain and electric field. C) The description for "maximum current" being defined as at the resonant frequency is misleading. Most common piezoelectric materials, including PZT as utilized by the authors, have mechanical hysteresis that results in significant loss and thermal challenges before the current level described by the authors can be reached. Further, most piezoelectric components have very high characteristic impedance, so they perform best at high voltage, low current operating points. Since improving current density is a significant focus of this paper, it is important that these descriptions of the current limits are accurate.

4. While the proposed circuit-level technique appears to be useful for improving current density, it doesn't tell the whole story. Using a piezoelectric resonator with a very high ratio between its diameter and thickness reduces its characteristic impedance, allowing it to operate with higher current at lower voltages with high efficiency. The proposed experiment uses a piezo with a very high diameter/thickness ratio, so this nuance should also be explained.

5. Since the authors have considered transistors, gate drivers, etc. in the density measurement, it is difficult to differentiate how much of the current density improvement is from the proposed circuit technique vs. everything being integrated onto an IC. Most of the highest-density piezo designs to date such as [46] (which should be added to both parts of Fig. 6) and [47] have been implemented with discrete switches, gate drivers, etc., and it is to be expected that further gain in converter density would be possible through integrating switches and gate drivers onto an IC. For a more representative density comparison, the authors should consider the converter's full output current (or power) and only the piezoelectric device's size, which is the actual bottleneck and an open research question.

6. It is not clear why the authors chose to compare volumetric current density (A/cm^3). More representative units in power conversion are areal current density (A/cm^2) and volumetric power handling density (W/cm^3). However, the proposed approach does not seem to perform as outstandingly based on these density metrics, especially when focusing on the piezo's size as recommended above for a fair evaluation of the proposed approach.

Reviewer #3

(Remarks to the Author)

All the comments I raised previously have been well addressed by the authors. I am fully satisfied with their revisions and believe the manuscript is now ready for publication.

Reviewer #6

(Remarks to the Author)

This paper proposes a new PR-based converter. The experimental validation is thorough, and the comparison with state-of-the-art designs shows clear advantages in current density and VCR capabilities. I have the following comments.

1. The introduction jumps between concepts without establishing clear logical flow. Consider restructuring to present the problem, existing limitations, and proposed solution more systematically.
 2. While the paper claims $K=1$ utilization factor at optimal conditions, the derivation of key equations (particularly the relationship between EFC voltage and optimal VCR) lacks sufficient detail.
 3. The proposed topology requires 13 power switches and 5 flying capacitors compared to simpler baseline designs. No analysis is provided regarding the cost-benefit tradeoff or realistic implementation complexity in production systems.
 4. Finally, an open question. This article mainly focuses on the converter itself, which is an open-loop control. However, in many industrial applications, closed-loop control is of concern. Please add relevant discussions.
- I also evaluated the authors' response to Reviewer #4. I believe the authors have adequately addressed the technical issues raised in the previous review.

Version 3:

Reviewer comments:

Reviewer #1

(Remarks to the Author)

1. Contrary to what the authors suggest, volumetric current density (A/cm^3) is not the most relevant metric for comparison. More representative (and widely utilized) metrics for power conversion are areal current density (A/cm^2) and volumetric power handling density (W/cm^3). This is supported by all of the works the authors cite for comparison in Fig. 7, including works from the authors' same research group. It appears the authors have elected to use volumetric current density because it makes this work appear more outstanding, rather than metrics that are most commonly adopted and useful. The primary density metric for all discussion and comparison on page 8 should be either areal current density (A/cm^2) or volumetric power handling density (W/cm^3).

2. Once the authors present the experimental results more fairly using the areal current density (A/cm^2) metric, the experimental results do not as strongly support the claimed benefits of the proposed technique. Most of the highest-density piezo designs to date have been implemented with discrete switches, gate drivers, etc., and it is to be expected that further gain in converter density would be possible through integrating switches and gate drivers onto an IC. So the most fair comparison would focus on only the piezoelectric device's size for areal current density, which is the actual bottleneck and an open research question. The authors provide this comparison in Fig. 7d. What is notably shown in Fig. 7d is that even

when the entire converter output current is considered for the proposed work, including the current that would be traveling through the AMP and not the piezo, the total current density with respect to piezo size is not greater than a baseline converter using the piezo to process all of the current. This comparison is quite fair since the PR cell operates with $K=1$ and the same voltage swing as the baseline converter (i.e., the VCR of the whole converter itself (the y axis of Fig. 7d) is not as relevant to the PR current density comparison). Thus, the experimental results do not strongly support the claimed benefit and impact.

3. There are still multiple inaccurate descriptions. (a) The abstract says “baseline PR-based dc-dc topologies have a fixed optimal conversion ratio”, which is a bit misleading. Baseline PR-based topologies achieve flat maximum efficiency during the continuous range of $1/2 < V_{out}/V_{in} < 1$, and this efficiency drops off continuously below $1/2$. This is different than a “fixed” conversion ratio as in switched capacitor converters, so this description should be modified accordingly. (b) Page 2 says “the high coupling and quality factor ($K \times Q$)”, but it’s actually $k^2 \times Q$ that is needed for high efficiency as evident in multiple cited works. If the authors are using K to denote a function of k^2 such as $k^2/(1-k^2)$, they should define that. (c) Page 2 says “PRs tend to exhibit optimal performance at the resonant frequency within the inductive region”, but impedance is purely real at the resonant frequency and is only inductive above the resonant frequency. This should instead read “near the resonant frequency”. (d) Perhaps the more relevant PZT comparison to lithium niobate would be using [52] rather than [28].

Reviewer #6

(Remarks to the Author)

All my concerns have been solved, the manuscript can be accepted now.

NCOMMS-24-85601-T - Response to Reviewers Comments

Title: A Hybrid Piezoelectric-Resonator-based DC-DC Converter

Authors: Jae-Young Ko, Wen-Chin B. Liu, and Patrick P. Mercier

Thank you for taking the time for our paper and providing useful feedback - we truly appreciate it. You will find our responses below in blue, bolded font.

Reviewer #1

Comments to the Author

This reviewer sees the most noteworthy contribution of this paper as being the use of “embedded flying capacitors” in piezoelectric resonator-based converters to enable higher voltage conversion ratios with high efficiency. Other aspects of this work (hybridizing piezoelectric-resonator-based converters with switched capacitor networks, implementing power converter circuits and control on an IC, etc.) have been previously investigated. This reviewer has the following concerns:

Thank you for your kind remarks! We appreciate you taking the time to review the paper carefully and provide good constructive feedback. We’d like to take this opportunity to acknowledge that yes, we agree the embedded flying capacitor technique is the most noteworthy contribution of the proposed system, though we also feel that the always-multi-path concept – which hadn’t been applied to piezoelectric resonator-based DC-DC converters – also represents an important contribution to further improve the performance of the design relative to state-of-the-art. That, coupled with the fact that only a single other publication has explored implementing power converter circuits for such an application on IC, and in this work we’ve carefully engineered the design to operate at much higher voltages, current density, and efficiency, overall represents a big leap in performance that we are excited to disseminate to the community.

1. There are some inaccuracies and incorrect terminology in the introductory descriptions of piezoelectric components. Line 50 - the provided definitions for C_p and C are incorrect. C_p is the static capacitance (i.e., the portion of the parallel plate capacitance that does not participate in mechanical resonance), and C is a modeled mechanical (sometimes referred to as motional) capacitance. Both combine together to form the parallel-plate capacitance at dc. Line 63 - the term "saturation current" is not applicable to PRs, so it is unclear what the authors mean by this in a physical sense.

Thank you for your comment. We have clarified what the terminology for the capacitors. Regarding saturation – what we mean by this is that the PRs can’t effectively operate beyond some internal current limit, which is akin to what happens in an inductor, though via a different

NCOMMS-24-85601-T - Response to Reviewers Comments

Title: A Hybrid Piezoelectric-Resonator-based DC-DC Converter

Authors: Jae-Young Ko, Wen-Chin B. Liu, and Patrick P. Mercier

mechanism. We've replaced the word "saturation current" with "maximum PR current" to make this more clear.

2. The authors claim that some things have not been explored before that actually have. Line 190 - the authors report that "no work has reported on the output voltage ripple of PR-based DC-DC converters," but this was explored via interleaving in "A Spurious-Free Piezoelectric Resonator Based 3.2 kW DC-DC Converter for EV On-Board Chargers" by Stolt et al.

Thank you for the comment and for pointing this out. We've updated the text to describe that work and modified the line you mentioned accordingly.

3. The comparison provided in Fig. 6 is a bit tenuous, or perhaps the authors' parameters for the comparison are not clear. While it's true that [24] does not have a high VCR or current density, its peak efficiency (99% in Figs. 24, 27, and 28) is misrepresented. For the current density comparison, there have been more recent designs of the baseline topology with greater density such as "A Piezoelectric-Resonator-Based DC-DC Converter Demonstrating 1 kW/cm³ Resonator Power Density" by Boles et al.

Thank you for your comment. The efficiency shown in Figure 6 corresponds to the peak efficiency achieved in the operating region where the PR terminals are connected in the sequence of V_{IN} - V_{OUT} , Zero, and V_{OUT} , where $V_{IN} > 2V_{OUT}$. Since our goal is to achieve high VCR with high efficiency, we compare the topology from the enumeration paper [24] that demonstrated the highest VCR and operated with the same switching sequence of V_{IN} - V_{OUT} , Zero, and V_{OUT} , where $V_{IN} > 2V_{OUT}$, as used in other works²⁹⁻³². We have added this information to manuscript.

Thank you also for point out that paper – we've gone ahead and added it to the comparison table. When calculating the current density, we included not only the piezoelectric resonator volume, but also all components that is necessary to regulate output voltage such as power switches, diodes, gate drivers, and capacitors. Therefore, the density value in our comparison table may differ from the numbers shown in the reference papers. We have noted this description and detailed number to the comparison table in Figure 6(a).

4. Some "for the first time" claims are a bit oversold. For example, the net optimal VCR of 9:1 is only a small improvement over previous designs with optimal VCRs of 8:1, and the methods used in those designs are easily extendable to higher conversion ratios.

NCOMMS-24-85601-T - Response to Reviewers Comments

Title: A Hybrid Piezoelectric-Resonator-based DC-DC Converter

Authors: Jae-Young Ko, Wen-Chin B. Liu, and Patrick P. Mercier

Thank you for your comment. We agree that 9:1 itself is only a modest improvement over 8:1, though overall it does come with multiple benefits. Importantly, extending from 8:1 to a higher conversion ratio based on prior topologies would require a higher-order switched-capacitor network, which may have further losses, and doesn't offer the embedded flying capacitor benefits the proposed topology does. With all that said, we've dialed back some of the language in the text as a result of this comment.

5. Even though it is very thin, the piezoelectric resonator seems to take up a lot of surface area, so it is unclear how practical this approach actually is without significant development of piezoelectric resonator itself. The authors do not seem to have innovated on that aspect of the proposed system, yet it seems like the largest bottleneck.

Thank you for the comment. We agree that piezoelectric resonator-based DC-DC converters are heavily influenced by piezoelectric resonator itself. But indeed, this is not the primary scope of the proposed research – we specifically used a commercially available device to demonstrate how the proposed topological advances can benefit piezoelectric-resonator-based converters as a whole, all while allowing for a more apples-to-apples quantitative comparison to prior art that, in many cases, used the same resonator we did. Since PR-based converters is still a nascent field with only a few published works, innovations in topology and circuit design can represent a major advance in the field. In our case, despite only using a commercially available resonator, the innovations in topology and chip design offers a major advance in performance relative to state of the art for applications targeting large conversion ratios such as data center power delivery, which is why we are exciting to share these results with the community, who can then build upon these results to further advance the field. Indeed, some of our next steps in research aim to investigate new materials and packaging strategies. We aim to explore this over the coming years in future publications. We tried to capture the excitement about the scaling potential of PRs in the introduction of the paper to better reflect the overall status of the field.

NCOMMS-24-85601-T - Response to Reviewers Comments

Title: A Hybrid Piezoelectric-Resonator-based DC-DC Converter

Authors: Jae-Young Ko, Wen-Chin B. Liu, and Patrick P. Mercier

Reviewer #2

Comments to the Author

Authors propose a modified version of the baseline piezoelectric-resonator-based DC-DC converter which makes it hybrid and efficient thanks to the introduction of the always-multi-paths with backside SC network.

The novelty degree is sufficient for the publishing of the work. The paper is well-written and structured and technically sound. The noteworthy results rely on the novel approach using the embedded flying capacitor (EFC), which allows to shift the VCR to achieve the maximum charge transfer utilizing factor k , and the use of the always-multi-path for balanced energy transfer in the circuit. The experimental results are in-line with the carried out analysis and comparison with the state of the art is provided.

Thank you for your kind remarks! We appreciate you taking the time to review the paper carefully and provide good constructive feedback.

My only one comment is about the comparison with the state of the art which require, in my opinion, more works. Further, the number of external devices should be reported in the comparison. How did the Authors calculate the current density for discrete component solutions? The comparison by using this metric results too far lookint at the compared works. In conclusion, comparison with prior art, which is very important for the proposal of the work, should be better stated.

Thank you for the comments. When we calculate the current density, we included not only the piezoelectric resonator volume, but also all components that is necessary to regulate output voltage such as power switches, diodes, gate drivers, and capacitors. We have added detailed number of this in Figure 6 comparison table, and we also added several rows such as separate VCR of PR and SC network, inductive region of PR, and additional notation to provide a clearer comparison.

Reviewer #3

Comments to the Author

This manuscript demonstrates a groundbreaking integrated circuit for piezoelectric resonator (PR)-based DC-DC conversion operating at a 48V input—remarkably achieving state-of-the-art efficiency and current/power density. Such high-voltage capability is a pioneering result that will likely open exciting new directions in next-generation data center power conversion. The paper is well structured, explaining both the Embedded Flying Capacitor (EFC) method and the Always Multi-Path (AMP) topology in a clear, logical manner. Given its strong experimental validation and its potential to advance the field significantly, I highly recommend this manuscript for publication

NCOMMS-24-85601-T - Response to Reviewers Comments

Title: A Hybrid Piezoelectric-Resonator-based DC-DC Converter

Authors: Jae-Young Ko, Wen-Chin B. Liu, and Patrick P. Mercier

in Nature Communications. Although the current draft is well-prepared, I would like to highlight a few minor comments below. I hope you consider these as constructive suggestions to further improve the paper and make your valuable contribution even more impactful.

Thank you for your kind remarks! We appreciate you taking the time to review the paper carefully and provide good constructive feedback.

1. If there is sufficient room in the manuscript, consider adding a concise figure or table summarizing representative PR-based converter configurations. For instance, show how previous studies have used frontside switched-capacitor (SC), backside SC, or cascaded stages, then highlight each configuration's inherent optimal voltage conversion ratio (VCR) and limitations. Such a quick visual comparison would help readers grasp precisely how your EFC + AMP approach diverges from and improves upon existing methods. For example,

[28] Two-stage: PR-based converter (2:1) + SC converter (2:1)

[32] Single-stage: Frontside SC (2:1) + PR conversion (2:1) + Backside SC (2:1)

[This work] Single-stage: EFC-based PR conversion (3:1) + Backside SC (3:1)

Thank you for the comment. We have added several lines to the comparison table in Figure 6 to enable a clear comparison across topologies. We also added the following figure to the supplementary figures section to further generalize the state-of-the-art landscape in relation to the proposed approach.

NCOMMS-24-85601-T - Response to Reviewers Comments

Title: A Hybrid Piezoelectric-Resonator-based DC-DC Converter

Authors: Jae-Young Ko, Wen-Chin B. Liu, and Patrick P. Mercier

[24]: (a) with single-stage, PR stage (opt @ 2:1)

[28]: (c) with two-stage, PR stage (opt @ 2:1) + Backside SC (4:1)

[30]: (b) with single-stage, Frontside SC (2:1) + PR network (opt @ 2:1)

[31]: (c) with single-stage, PR network (opt @ 2:1) + Backside SC (2:1) with partial-dual-path

[32]: (d) with single-stage, Frontside SC (2:1) + PR network (opt @ 2:1) + Backside SC (2:1) with partial-dual-path

This work: (e) with single-stage, PR network with EFC (opt @ 3:1) + Backside SC (3:1) with always-multi-paths

2. In the early theoretical sections, the manuscript states that the EFC voltage is set to a multiple of the output voltage to shift the PR's inherent 2:1 ratio to a higher value (e.g., 3:1). Clarifying why this choice (as opposed to another reference voltage) is crucial to realizing the $K=1$ condition would further strengthen the reader's understanding of the design trade-offs.

Thank you for the comment. Although it is possible to configure the EFC voltage to different levels, it is important to consider the EFC's discharging path without interrupting powering stage for achieving $K=1$. As shown in Figure 2(b), the EFC is conceptually discharged during phase 4 and charged during phase 6. Since phase 6 is generally longer or the same compared to phase 4 ($q_6 \geq q_4$), it requires an additional path for the charge balance of the EFC.

NCOMMS-24-85601-T - Response to Reviewers Comments

Title: A Hybrid Piezoelectric-Resonator-based DC-DC Converter

Authors: Jae-Young Ko, Wen-Chin B. Liu, and Patrick P. Mercier

When the EFC voltage is set to a multiple of the output voltage, charge balance can be maintained using a simple 1:1 SC network composed of CF1 and CF2. In this configuration, the entire energy stored in the EFC can be easily delivered to the output without interrupting powering stage, thereby allowing the system to maintain $K=1$. We've added this explanation to manuscript to highlight the motivation behind our design choice.

3. While the innovation emphasizes large step-down (from 48V to a few volts), it would be helpful to briefly mention the highest feasible output voltage (e.g., under a 48V input) that still maintains the proposed benefits—such as soft-switching, high efficiency, and $K=1$.

Thank you for the comment. The maximum output voltage of AMP-EFC converter is theoretically one-ninth of the input voltage. With a target input voltage of 48V, this corresponds to a maximum achievable output voltage is 5.33V. This allows full regulation across the commonly used nominal voltage range below 5V. As shown in Figure 5(d), the measured efficiency improves as the VCR approaches 9, due to an increase in the utilization factor toward unity. In our current implementation, switches S9-S13 are designed using 5V-rated devices. As this constraint limits the measurement range to below 5V, we conducted measurement up to 4.8V (VCR=10). However, without this device constraint, we expect that even higher efficiency than the currently measured peak of 96.2% could be achieved at >5V output voltage. Furthermore, the switches annotated as operating under ZVS in Figure 4(a) consistently exhibit soft-switching behavior across all VCR conditions.

4. You note that AMP improves current density by distributing current among multiple paths and reducing PR current amplitude. Another advantage likely includes reduced output ripple via more continuous energy delivery. Please consider providing a comparison—either numeric or waveform-based—showing how AMP specifically lowers output ripple compared to standard single- or partial-path approaches.

Thank you for the comment. Unlike PR current, which can be approximated as a sinusoidal waveform, the capacitor current in the SC network is difficult to express analytically. Therefore, it is challenging to calculate the voltage ripple with exact numerical precision. However, it can be reasonably estimated based on the amount of charge transferred by the capacitors.

For example, when the PR current is positive (phases 1-4) and when it is negative (phases 5-7), each condition corresponds to a half-cycle. The smaller the difference in transferred charge

NCOMMS-24-85601-T - Response to Reviewers Comments

Title: A Hybrid Piezoelectric-Resonator-based DC-DC Converter

Authors: Jae-Young Ko, Wen-Chin B. Liu, and Patrick P. Mercier

between these two intervals, the more uniformly energy is delivered over a full cycle. Using the equation shown below, we analyzed how the charge transfer is distributed in partial dual-path configurations such as those presented in [31] and [32].

$$Q_{OUT} = Q_{PR} + Q_{CF} = 2|q_2| + 2|q_6| = P_{OUT} / f_{sw} V_{OUT}$$

$$Q_{PR} = |q_2| + |q_6| = Q_{OUT} / 2$$

$$* q_2(V_{IN,PR} - V_{O,PR}) + q_6(V_{O,PR}) = 0$$

The architecture proposed in [32] benefits from the assistance of a frontside SC network, resulting in a higher utilization factor K at high VCRs compared to [31]. This suggests improved current distribution at high VCRs, even though both [31] and [32] employ the same partial dual-path configuration. However, the structure in [32] only operates in a dual-path during phase 6, during which more than 80% of the total energy per cycle is transferred, leading to an unbalanced current path. We performed the same analysis on the proposed AMP-EFC converter, and the detailed calculations are provided in Supplementary Figure 2.

NCOMMS-24-85601-T - Response to Reviewers Comments

Title: A Hybrid Piezoelectric-Resonator-based DC-DC Converter

Authors: Jae-Young Ko, Wen-Chin B. Liu, and Patrick P. Mercier

On the other hand, the proposed AMP-EFC converter operates in a multi-path configuration during phase 2 and phase 6. Compared to the topology in [32], the AMP-EFC demonstrates a significantly more balanced current distribution. To support the validity of our charge-based analysis, we conducted output waveform simulations for the baseline PR converter [24], DSPPR [32], and the proposed AMP-EFC converter, shown below. We have included a brief summary of this analysis in the paper, and included the simulation plot below as Supplementary Figure 4.

Same condition: $V_{IN} = 48V$, $V_{OUT} = 5V$, $I_{OUT} = 200mA$, $C_{OUT} = 10\mu F$

NCOMMS-24-85601-T - Response to Reviewers Comments

Title: A Hybrid Piezoelectric-Resonator-based DC-DC Converter

Authors: Jae-Young Ko, Wen-Chin B. Liu, and Patrick P. Mercier

5. The text states that all switches are zero-voltage switched, but lines 226 and 238 raise questions about whether S8, S11, S12, and S13 always meet ZVS criteria. Including an explicit on/off timing diagram or notation in Figure 4(b) confirming which switches are ZVS-enabled—and at which phase transitions—would enhance the credibility of the soft-switching claim.

Thank you for the comment. In our topology, switches S8, S11, S12, and S13 do not operate with totally zero voltage switching. Since the nodes VX6 and VX7 across CF5 behave inversely to the rising/falling of VX4 and VX5, respectively, completely soft switching cannot be achieved for these 4 switches. Fortunately, the drain-source voltage (VDS) of these switches is limited to VOUT (<5V in our target), which is significantly lower than that of other switches operating at tens of voltage. As a result, the impact of hard switching on these switches is not considered critical in terms of either reliability or power loss. We have annotated the switches that are capable of operating with ZVS in Figure 4 and also modified the manuscript when discussing switching losses to make things more clear.

6. One key advantage noted is the minimized normalized PR current. As lower resonator current could mitigate vibration losses, reduce temperature rise, and improve reliability, please elaborate on how these improvements translate into higher output power capability or longevity. This will reinforce the importance of suppressing PR current amplitude. I guess that lowering the PR current amplitude correlates with higher maximum output current. Briefly elucidating the mechanism (e.g., PR saturation or Q-factor constraints) that dictates this upper bound on output current would strengthen the understanding of the design trade-offs.

Thanks for the comment. That is a good point, and your assumption that the PR current is directly related to the maximum output current is indeed correct. PRs exhibit different BVD models and inductive regions depending on their size and material properties. The measurement and modeling of the PR used in our work are provided in Supplementary Figure 1. Since a PR-based DC-DC converter delivers the highest current when operating at the lowest frequency within the inductive region, the upper bound of the energy that a PR can transfer is ultimately limited by this condition. Thus, the purpose of the circuits that augment the baseline PR-based DC-DC converter topology essentially strive to reduce the current the PR needs to process by providing additional voltage step-down capabilities via switched-capacitor circuits, ultimately increasing the total output current the structure can process. Accordingly, we have added a description of this limitation in the Introduction section of the manuscript.

NCOMMS-24-85601-T - Response to Reviewers Comments

Title: A Hybrid Piezoelectric-Resonator-based DC-DC Converter

Authors: Jae-Young Ko, Wen-Chin B. Liu, and Patrick P. Mercier

7. Verify the labels on the horizontal and vertical axes. The intended label seems to be “VCR = 10” (not 0.1) for the x-axis, and “Output Current [A]” (not [mA]) for the y-axis. Correcting these minor issues will ensure clarity in your otherwise excellent measurement results.

Thank you for the comment. We have modified the labels.

8. To strengthen the work’s real-world relevance—especially for data centers—compare your results against at least one state-of-the-art inductor-based solution operating in the same high-voltage, high-step-down domain. Demonstrating that a PR-based converter can rival or outperform existing inductive legacy solutions (in terms of efficiency or volumetric density) will highlight the practical significance of this novel PR-based topology.

Thank you for the comment. While we agree this would be useful in general, it turns out it’s quite difficult to do an apples to apples comparison across different energy storage elements. It should also be pointed that that inductors have had many decades to improve in performance, and yet no commercially available PRs have been optimized for power applications yet, as this is a nascent field. The primary purpose of this paper is to improve upon existing PR-based converters by proposing new a topology that is compatible with various PR devices, thereby enabling enhanced performance and/or functionality, with overall performance improving further with resonator improvements. Therefore, we believe that the comparison of our work to other studies using PR devices is more appropriate and relevant to the aims of this work.

9. In Figure 6, please clarify how you compute the converter volume—e.g., whether it includes the PR device’s thickness, external capacitors, IC packaging, etc. Explaining these details will help readers fairly compare your density metrics with those in other works.

Thanks for your comment. When we calculate total volume, we included all components that is necessary to regulate output voltage such as piezoelectric resonator device, power switches, diodes, gate drivers, and capacitors. We have added detailed number included in our calculation in Figure 6 comparison table.

10. The paper employs an external DSP to run the converter under open-loop control. Since this circuit appears fully capable of closed-loop regulation, it would be helpful to clarify why you chose an open-loop scheme for validation. Was the DSP approach simply more convenient for rapid prototyping or parameter sweeping? Addressing this question will assure readers that closed-loop control is indeed feasible and that open-loop operation was selected primarily to streamline experimental demonstration.

NCOMMS-24-85601-T - Response to Reviewers Comments

Title: A Hybrid Piezoelectric-Resonator-based DC-DC Converter

Authors: Jae-Young Ko, Wen-Chin B. Liu, and Patrick P. Mercier

Thank you for the comment. Yes, you are right - to enable a flexible and simplified parameter sweep, the experiments were conducted using an open-loop configuration based on a DSP. Since our topology employs a merged SC network, the SC operates in synchronization with the control signals of the PR network. Even though the PR network includes an EFC, it maintains compatibility with the baseline PR-based DC-DC converter in terms of control, as the operation is still divided into seven phases. The main purpose of this paper was to demonstrate an advance in topology and how this could impact overall system performance – which we feel we have. Given, however, that the system uses similar driving waveforms to other topologies, the system could readily use existing closed-loop control strategies (for which there are exceedingly few in the published literature) without requiring major modifications. This, however, is out of scope of the present manuscript. We've updated the manuscript to reflect that the proposed topology would be compatible with existing control techniques.

Reviewer #4

Comments to the Author

This manuscript presents a new circuit topology for a piezoelectric-resonator-based dc-dc converter. This circuit topology is derived by introducing an embedded flying capacitor (EFC) within the structure of a well-established circuit topology for piezoelectric-resonator-based dc-dc converters. The concept of introducing the EFC is similar, though not identical, to the concept of introducing a series capacitor in a buck converter to form the series capacitor buck converter. The EFC enables the proposed dc-dc converter topology to reduce the amount of charge that needs to be circulated within the piezoelectric-resonator at larger voltage conversion ratios. This is the main innovation presented in this manuscript. The authors also present an implementation of the proposed converter and its measured performance. Below are specific comments and questions for the authors:

Thank you for your kind remarks! We appreciate you taking the time to review the paper carefully and provide good constructive feedback. We'd like to take this opportunity to acknowledge that yes, we agree the embedded flying capacitor technique is the most noteworthy contribution of the proposed system, though we also feel that the always-multi-

NCOMMS-24-85601-T - Response to Reviewers Comments

Title: A Hybrid Piezoelectric-Resonator-based DC-DC Converter

Authors: Jae-Young Ko, Wen-Chin B. Liu, and Patrick P. Mercier

path concept – which hadn't been applied to piezoelectric resonator-based DC-DC converters – also represents an important contribution to further improve the performance of the design relative to state-of-the-art. That, coupled with the fact that only a single other publication has explored implementing power converter circuits for such an application on IC, and in this work we've carefully engineered the design to operate at much higher voltages, current density, and efficiency, overall represents a big leap in performance that we are excited to disseminate to the community.

1. Given that the introduction of the EFC is the main innovation of this manuscript, the authors should explain more clearly, including intuitively, how the presence of the EFC achieves a reduction in charge circulation at larger voltage conversion ratios. The current explanation related to the optimal voltage conversion ratio (VCR) for the conventional piezoelectric-resonator (PR) converter and the proposed PR converter provided in the first two paragraphs of the section titled "Embedded flying capacitors for modifying the PR network's optimal VCR" is not clear.

Thank you for the comment. In the manuscript, we have attempted to explain the EFC using a switching-sequence-based approach, consistent with the sequences commonly used in conventional PR converters. In conventional PR converters, the voltage across the PR terminals, V_{CP} , follows a switching sequence of $V_{IN,PR}$, $V_{OUT,PR}$, Zero, and $V_{OUT,PR}$. This means the PR's static capacitance (CP) should be fully discharging for circulating phase (normally phase 4). However, the EFC creates a new DC node (other than $V_{IN,PR}$ and $V_{OUT,PR}$) with the DC voltage of the EFC, V_{EFC} . With this new DC node, the V_{X1} node is no longer directly connected to $V_{OUT,PR}$, and operates with new switching sequence of $V_{IN,PR}$, $V_{OUT,PR}$, V_{EFC} , $V_{EFC}+V_{OUT,PR}$, from which a new optimal VCR can be derived: $V_{IN,PR}-V_{OUT,PR}=V_{EFC}+V_{OUT,PR}$. In other words, the EFC reduces charge circulation by making it unnecessary for the PR to be fully discharging. In addition to the original description of the EFC concept, we've added this analysis to clarify, more intuitively, how the EFC concept works and how it serves to further improve converter performance.

2. The authors should also discuss any limitations or constraints imposed on the design or operation of the PR converter by the introduction of the EFC.

Thank you for the comment. When introducing the EFC scheme, it is important to consider the discharging path. As shown in Figure 2(b), the EFC is conceptually discharged in phase 4 and charged in phase 6. Since phase 6 is generally longer or same compared to phase

NCOMMS-24-85601-T - Response to Reviewers Comments

Title: A Hybrid Piezoelectric-Resonator-based DC-DC Converter

Authors: Jae-Young Ko, Wen-Chin B. Liu, and Patrick P. Mercier

4 ($q_6 > q_4$), it requires an additional path for the charge balance of the EFC. Since the key goal of our design is to ensure that the discharging path is implemented without any decrease in the K, we set the EFC voltage to a multiple of the output voltage. This setting provides simple a simple solution for EFC charge balance without additional control circuit. We believe that adding this explanation to manuscript will not only clarify the motivation behind our design choice, but also alleviate potential concerns that it may impose a design constraint.

3. The authors should also describe how the required value of the voltage across the embedded flying capacitor is determined for a given input voltage, output voltage and any other parameters of the proposed PR converter.

Thank you for the comment. In our design, the EFC voltage is influenced solely by the output voltage. The switching of CF1 and CF2 ensures that the EFC voltage remains self-balanced and aligned with $V_{OUT,PR}$ ($=3V_{OUT}$), as shown in Figure 3(b). Furthermore, due to the 3:1 backside SC network, the EFC voltage is set to $V_{EFC} = V_{OUT,PR} = 3V_{OUT}$. We noted this voltage in Figure 4(a).

4. The authors should describe which components are included in the calculation of current density, and comparison with the state-of-the-art, in Fig. 6. For example, is the area of the piezoelectric resonator and the other passive components included in this calculation and comparison.

Thank you for the comment. When we calculate the current density, we included all components that is necessary to regulate output voltage such as piezoelectric resonator device, power switches, diodes, gate drivers, and capacitors. We have added volume of piezoelectric resonator and other components included in our calculation to the table in Figure 6.

5. The authors should describe the thermal management strategy utilized for the converter during its testing. The authors should also comment on the factors limiting the testing of the converter at power, or current, levels higher than those at which the converter was tested.

Thank you for the comment. We tested the proposed converter in a normal open air lab environment, and did not include any special thermal management techniques (e.g., heatsinks). Presumably performance, including output current, power, and efficiency, could increase with more thermal management employed.

NCOMMS-24-85601-T - Response to Reviewers Comments

Title: A Hybrid Piezoelectric-Resonator-based DC-DC Converter

Authors: Jae-Young Ko, Wen-Chin B. Liu, and Patrick P. Mercier

The output voltage of our converter is limited to 5V due to the use of 5V-rated transistors (S9-S13) at the output stage of the fabricated IC. The current limitation is determined by the PR device characteristics. Since PR-based converter fail to operate properly when the switching frequency exceeds the inductive region of the PR, the maximum deliverable current is defined at the lowest frequency within this region. We refer to this value as the maximum PR current, as defined in the manuscript. Under a 48V input, the maximum output current is 1A. Consequently, the maximum output power of the AMP-EFC converter is 5W. We have added this voltage limit to main discussion.

6. The authors note that one strategy currently being employed or considered for 48 V to point-of-load conversion in applications such as data centers is the use of hybrid magnetic switched capacitor converters. The authors should also compare the performance (power density and efficiency) of their proposed hybrid piezoelectric-resonator converter with the state-of-the-art hybrid magnetic switched capacitor converters.

Thank you for the comment. While we agree this would be useful in general, it turns out it's quite difficult to do an apples to apples comparison across different energy storage elements. It should also be pointed that that inductors have had many decades to improve in performance, and yet no commercially available PRs have been optimized for power applications yet, as this is a nascent field. The primary purpose of this paper is to improve upon existing PR-based converters by proposing new a topology that is compatible with various PR devices, thereby enabling enhanced performance and/or functionality, with overall performance improving further with resonator improvements. Therefore, we believe that the comparison of our work to other studies using PR devices is more appropriate and relevant to the aims of this work.

7. Given that the main application the proposed converter is targeting is data centers, and the power (and output current) requirement of data center point-of-load converters is increasing (and in some cases already around 1000 A), the authors should discuss the scalability of the proposed converter to such current levels, and how the scaling laws of hybrid piezoelectric-resonator converters compare with the scaling laws of hybrid magnetic switched capacitor converters.

Thank you for the comment. This is a good point in favor of PR-based converters down the road – they scale very nicely compared to inductors. To get there, we'll need advances in PR-based materials and packaging, and operate such resonators at higher frequencies. We've

NCOMMS-24-85601-T - Response to Reviewers Comments

Title: A Hybrid Piezoelectric-Resonator-based DC-DC Converter

Authors: Jae-Young Ko, Wen-Chin B. Liu, and Patrick P. Mercier

mentioned these favorable scaling properties in the introduction section, and updated the conclusion section with some forward-looking opportunities in future work based on scaling.

Reviewer #5

Comments to the Author

Thank you for your efforts!

NCOMMS-24-85601-T - Response to Reviewers Comments

Title: A Hybrid Piezoelectric-Resonator-based DC-DC Converter

Authors: Jae-Young Ko, Wen-Chin B. Liu, and Patrick P. Mercier

Thank you for taking the time to read our paper and providing useful feedback - we truly appreciate it. You will find our responses below in blue, bolded font. Note that additions and major changes to the paper manuscript have been highlighted.

Reviewer #1

Comments to the Author

The revised version of the manuscript is improved but still has significant issues.

1. This work claims to be a solution for data center environments, but 48-5V (or 48-1V) data center power delivery tends to require kW (and often kA) power levels. The solution presented in this work is 2-3 orders of magnitude lower than this, so it is difficult to justify as a solution for this application.

Thank you for the comment. It's true that magnetic-based converters, which have been developed and optimized over many decades, are currently the dominant solution for delivering kW-level power in such systems. However, the scaling potential of magnetic components is approaching its limits, and further improvements are difficult to come by. In contrast, piezoelectric resonators (PRs) have recently (over the last ~5 years) emerged as a promising alternative in power conversion due to their more favorable scaling characteristics. While PR-based converters are not yet able to match the absolute power levels achieved by mature magnetic solutions, there is growing research interest in PRs across circuit design, materials, and packaging - driven by their high potential. We have added this content in lines 57-59. In this work, we introduce a novel circuit topology that leverages the unique benefits of PRs to reach state-of-the-art circuit-level results. This represents an important step towards meeting the needs of data centers. While PR converters aren't there yet, that's the purpose of research – to move forward and advance the field. We have pointed out this in lines 88-93. Your point is well-received, however, and we've modified the introduction to make the objectives and aspirations of the work clearer and more grounded, directly mentioning the potential, but noting that the field isn't there yet.

2. There are multiple unclear descriptions and confusing terminology choices in the introduction. A) The authors mention that power distribution systems are transitioning from 12V to 48V, and that this allows for the delivery of higher current in a more compact form factor. However, transitioning to higher voltage should reduce current for the same power level, so this sequence of sentences should be clarified. B) It is unclear what the authors mean

NCOMMS-24-85601-T - Response to Reviewers Comments

Title: A Hybrid Piezoelectric-Resonator-based DC-DC Converter

Authors: Jae-Young Ko, Wen-Chin B. Liu, and Patrick P. Mercier

by “iso-performance” when referring to magnetic components. C) The term “energy conservation” should probably be “energy storage” when referring to piezoelectric components.

Thanks for your comment. A) The point we were trying to make is that a conventional system distributes power with a 12V rail, which, to meet the high output power demands, requires high input currents and large I_{IN}^2R losses. Moving to a 48V rail reduces the input current for the same output power level, leading to lower I_{IN}^2R losses. We’ve clarified this point in the manuscript at lines 25-27. B) Our intention was simply to note that the sub-linear scaling properties of magnetic components make it difficult to achieve further miniaturization while maintaining performance (lines 39-41). We have modified the writing there to be more clear. C) Agreed – we have modified this part (line 46).

3. There are multiple inaccuracies in the introduction that are concerning. A) There is no such thing as a physical turns ratio in piezoelectric transformers; as written, this sentence risks confusing readers. There is an ideal transformer in the model for a piezoelectric transformer, and there are many factors (material properties, geometric dimensions, etc.) that determine its transformation ratio. B) It is not clear what the authors mean by “beyond which they do not operate correctly” when referring to the maximum current of a piezoelectric component. Piezoelectric components do not have a maximum current rating; the relevant physical limits are mechanical stress/strain and electric field. C) The description for “maximum current” being defined as at the resonant frequency is misleading. Most common piezoelectric materials, including PZT as utilized by the authors, have mechanical hysteresis that results in significant loss and thermal challenges before the current level described by the authors can be reached. Further, most piezoelectric components have very high characteristic impedance, so they perform best at high voltage, low current operating points. Since improving current density is a significant focus of this paper, it is important that these descriptions of the current limits are accurate.

Thank you for the comments. A) We agree with your concern that the sentence may be confusing to readers. Since the transformation ratio of piezoelectric transformers (PTs) depends on various design parameters, a detailed explanation would require substantial discussion. Given that our work focuses on piezoelectric resonator-based operation, we chose to briefly mention the relevant characteristics of PTs in lines 53-55, in order to provide context without deviating from the main focus of the work. B) We have revised the text to clarify the discussion by explicitly listing the specific factors that determine the maximum current capability of the circuit in lines 77-79. C) We also agree that it’s important to clearly describe the current limitations, and that these are strongly influenced by device-level physical factors.

NCOMMS-24-85601-T - Response to Reviewers Comments

Title: A Hybrid Piezoelectric-Resonator-based DC-DC Converter

Authors: Jae-Young Ko, Wen-Chin B. Liu, and Patrick P. Mercier

We briefly addressed this point in lines 77-79, with a reference paper [16], where a more detailed device-level analysis is provided. Since our paper focuses on circuit-level solutions, we have expanded the discussion of circuit-side current limitations in lines 83-87.

4. While the proposed circuit-level technique appears to be useful for improving current density, it doesn't tell the whole story. Using a piezoelectric resonator with a very high ratio between its diameter and thickness reduces its characteristic impedance, allowing it to operate with higher current at lower voltages with high efficiency. The proposed experiment uses a piezo with a very high diameter/thickness ratio, so this nuance should also be explained.

Thanks for your comment. It's a good point, and we've made a note about this in lines 326-329. We'd also like to point out that multiple of the prior-art references, operating at similar input voltages (as listed in the table of comparisons), used resonators with higher diameter/thickness ratios than ours. But, nevertheless, it's good to be open and descriptive about device-level trade-offs, so your comment is appreciated, and the manuscript has been clarified and improved as a result.

5. Since the authors have considered transistors, gate drivers, etc. in the density measurement, it is difficult to differentiate how much of the current density improvement is from the proposed circuit technique vs. everything being integrated onto an IC. Most of the highest-density piezo designs to date such as [46] (which should be added to both parts of Fig. 6) and [47] have been implemented with discrete switches, gate drivers, etc., and it is to be expected that further gain in converter density would be possible through integrating switches and gate drivers onto an IC. For a more representative density comparison, the authors should consider the converter's full output current (or power) and only the piezoelectric device's size, which is the actual bottleneck and an open research question.

Thank you for the comment. It's a good suggestion, and we've gone ahead and included comparisons to current density based only on the size of the resonator itself as requested (both in the table of comparisons in Fig. 6 and in the new Fig. 7, which outlines plots of various different current density calculations). We would like to point out, however, that while yes, full integration can offer a size advantage, it's quite difficult to operate an IC at high voltages, which is where a discrete design actually may have some advantages. However, integration clearly

NCOMMS-24-85601-T - Response to Reviewers Comments

Title: A Hybrid Piezoelectric-Resonator-based DC-DC Converter

Authors: Jae-Young Ko, Wen-Chin B. Liu, and Patrick P. Mercier

offers advantages, as described in lines 391-393. With that said, we've included the data requested and pointed out the relative challenges with different implementation techniques in Figs. 6 and 7, ultimately allowing the reader to see the data and make their own conclusions for what is of highest interest to them. Regarding the comparison with [46], now renumbered as [50], we acknowledge that it's a great paper that reports very high current density. However, as you pointed out, making a fair assessment of circuit-level contributions requires comparing systems using piezoelectric devices of similar materials and operating conditions as now discussed in lines 353-357. All the work that we've compared to uses PZT and operates at 100-500kHz, whereas [50] uses LiNbO₃ and operates over 6MHz, precluding a reasonably apples-to-apples topological comparison (for example, [50] uses the same topology (1 cell case) as [28] and [52] which use PZT, and yet [50] achieves very different results). Instead, we've highlighted [50] and the use of LiNbO₃ as a materials-driven method in lines 402-405 to further improve current density in future work, complementary to the topological circuit innovations presented in this manuscript.

6. It is not clear why the authors chose to compare volumetric current density (A/cm^3). More representative units in power conversion are areal current density (A/cm^2) and volumetric power handling density (W/cm^3). However, the proposed approach does not seem to perform as outstandingly based on these density metrics, especially when focusing on the piezo's size as recommended above for a fair evaluation of the proposed approach.

Thanks for the comment. This Volumetric current density is commonly adopted in works targeting higher voltage conversion ratios (VCR) and high current capabilities, such as applications for data centers and automobiles. First, when comparing density metrics, it's important to distinguish between chip-level and system-level comparisons. Areal current density is typically used for chip-to-chip comparisons – for instance, in LDOs or integrated voltage regulators (IVR) – because the chip thickness are relatively similar and do not significantly affect performance. However, in system-level comparisons, like in our case with power converters, volumetric metrics are more appropriate. This is especially relevant when the geometry of PR device has a direct impact on the system's performance. In such cases, volumetric density provides a more accurate representation of capability. Second, regarding the use of current rather than power as a performance indicator, this is because prior-art reports results over very different voltage ranges, doing a power density comparison is not

NCOMMS-24-85601-T - Response to Reviewers Comments

Title: A Hybrid Piezoelectric-Resonator-based DC-DC Converter

Authors: Jae-Young Ko, Wen-Chin B. Liu, and Patrick P. Mercier

fair. Since our work targets high current delivery at high VCR (i.e., low output voltage), using power as a metric could be misleading as similar power values can result from either high current/low voltage or low current/high voltage scenarios. For example, under the same 48V input, a 5V/1A output represents a much larger (and more difficult to achieve) voltage conversion ratio, higher current stress on the PR device, and a better fit to say, for example, data center applications, compared to a 20V/0.25A output, even though both deliver the same output power. We have added this a discussion of our justification for use of current density to the manuscript in lines 357-364.

Nevertheless, for interested readers, we have also included the areal current density of this work in the comparison table (Fig.6) and in the comparison plots (now Fig. 7) as a reference, allowing readers to examine the data and draw their own conclusions based on their priorities.

Reviewer #3

Comments to the Author

All the comments I raised previously have been well addressed by the authors. I am fully satisfied with their revisions and believe the manuscript is now ready for publication.

Thank you for recognizing the value of our approach and findings! We appreciate you taking the time to review the paper carefully and provide constructive feedback.

Reviewer #6

Comments to the Author

This paper proposes a new PR-based converter. The experimental validation is thorough, and the comparison with state-of-the-art designs shows clear advantages in current density and VCR capabilities. I have the following comments.

1. The introduction jumps between concepts without establishing clear logical flow. Consider restructuring to present the problem, existing limitations, and proposed solution more systematically.

NCOMMS-24-85601-T - Response to Reviewers Comments

Title: A Hybrid Piezoelectric-Resonator-based DC-DC Converter

Authors: Jae-Young Ko, Wen-Chin B. Liu, and Patrick P. Mercier

Thanks for your comment. We've re-organized and re-written the introduction to flow more naturally and systematically, ultimately improving the clarity.

2. While the paper claims $K=1$ utilization factor at optimal conditions, the derivation of key equations (particularly the relationship between EFC voltage and optimal VCR) lacks sufficient detail.

Thank you for the comment. From the charge balance equation of capacitors C and C_p shown in supplementary Fig.3c), we can derive the relationship $q_2+q_4+q_6=0$. Considering that charge internally circulated ($q_4=0$) becomes zero at optimal conditions, it follows that $q_2=-q_6$. By substituting this into the energy balance equation of the PR, $q_2(V_{IN,PR}-V_{OUT,PR})+q_6V_{OUT,PR}=0$, we can derive the VCR of the baseline topology as 2. We have added this derivation in lines 177-180. Similarly, when the EFC is introduced, the same charge balance conditions holds, and the relationship $q_2=-q_6$ remains valid under optimal conditions. In this case, due to the new switching sequence, the PR's energy balance becomes $q_2(V_{IN,PR}-V_{OUT,PR})+q_6(V_{EFC}+V_{OUT,PR})=0$, from which we obtain the new optimal point of $2+V_{EFC}/V_{OUT,PR}$, as shown in line 192.

3. The proposed topology requires 13 power switches and 5 flying capacitors compared to simpler baseline designs. No analysis is provided regarding the cost-benefit tradeoff or realistic implementation complexity in production systems.

Thanks for your comment. This is always a nuanced point when discussing hybrid converters that incorporate additional switches and capacitors to improve the performance of an inductive (or now in this case a PR-based) converter. First, we'd like to point out that while the proposed topology does include more switches and capacitors compared to the baseline topology, it operates with the same 7-phase control scheme, does not require any additional circuitry for capacitor balancing, and still enables zero-voltage or zero-current switching. Therefore, the overall control complexity remains comparable to that of the baseline as now discussed in lines 397-399. Second, we'd like to point out that compared to the only other integrated-circuit-based implementation in [36], we use only 2 additional capacitors, and yet, even while including the volume of those capacitors (and the chip and PR volume), we achieve a 2.4x improvement in current density - we added this comment in lines 368-370. So as a result, there is a tangible current density advantage to the proposed technique. Regarding cost – yes,

NCOMMS-24-85601-T - Response to Reviewers Comments

Title: A Hybrid Piezoelectric-Resonator-based DC-DC Converter

Authors: Jae-Young Ko, Wen-Chin B. Liu, and Patrick P. Mercier

this will slightly increase cost to include a few more capacitors, but the small ceramic capacitors we use cost less than 1 US cent per unit at scale, so it shouldn't be a materially large cost, as discussed at lines 370-371. An exact estimate (including the overhead for a more complex PCB) is beyond the scope of this research. Nevertheless, we've included a brief discussion on this in lines 391-393.

4. Finally, an open question. This article mainly focuses on the converter itself, which is an open-loop control. However, in many industrial applications, closed-loop control is of concern. Please add relevant discussions.

Thank you for the comment. Indeed, our experiments were conducted using open-loop control to quickly evaluate the performance of the proposed topology. This is standard in the PR-based DC-DC converter literature. In fact, there's only a very small handful of papers in the published literature that demonstrate closed-loop control of PR-based converters. In other words, this is still an emerging topic with much work still to be done, and we have added this in lines 393-396. Nevertheless, even though the PR network in this work includes an EFC, it maintains compatibility with the baseline PR-based DC-DC converter in terms of control, as the operation is still divided into the same seven phases. Thus, the system could readily use existing closed-loop control strategies without requiring major modifications. We've mentioned a discussion about this in lines 397-399.

NCOMMS-24-85601-T - Response to Reviewers Comments

Title: A Hybrid Piezoelectric-Resonator-based DC-DC Converter

Authors: Jae-Young Ko, Wen-Chin B. Liu, and Patrick P. Mercier

Reviewer #1

Comments to the Author

1. Contrary to what the authors suggest, volumetric current density (A/cm^3) is not the most relevant metric for comparison. More representative (and widely utilized) metrics for power conversion are areal current density (A/cm^2) and volumetric power handling density (W/cm^3). This is supported by all of the works the authors cite for comparison in Fig. 7, including works from the authors' same research group. It appears the authors have elected to use volumetric current density because it makes this work appear more outstanding, rather than metrics that are most commonly adopted and useful. The primary density metric for all discussion and comparison on page 8 should be either areal current density (A/cm^2) or volumetric power handling density (W/cm^3).

Thanks for your comment. Per your request, we've now added volumetric power density to Figs. 6 and 7 (considering the volume of the PR alone, and the volume of all components), along with a related discussion in the text. In summary, we now have six density metrics featured in the paper: volumetric current density, volumetric power density, and areal current density – all considering the area/volume of the PR itself only, and the area/volume of the entire system – to give readers a broad span of metrics depending on their application needs.

2. Once the authors present the experimental results more fairly using the areal current density (A/cm^2) metric, the experimental results do not as strongly support the claimed benefits of the proposed technique. Most of the highest-density piezo designs to date have been implemented with discrete switches, gate drivers, etc., and it is to be expected that further gain in converter density would be possible through integrating switches and gate drivers onto an IC. So the most fair comparison would focus on only the piezoelectric device's size for areal current density, which is the actual bottleneck and an open research question. The authors provide this comparison in Fig. 7d. What is notably shown in Fig. 7d is that even when the entire converter output current is considered for the proposed work, including the current that would be traveling through the AMP and not the piezo, the total current density with respect to piezo size is not greater than a baseline converter using the piezo to process all of the current. This comparison is quite fair since the PR cell operates with $K=1$ and the same voltage swing as the baseline converter (i.e., the VCR of the whole converter itself (the y axis of Fig. 7d) is not as relevant to the PR current density comparison). Thus, the experimental results do not strongly support the claimed benefit and impact.

NCOMMS-24-85601-T - Response to Reviewers Comments

Title: A Hybrid Piezoelectric-Resonator-based DC-DC Converter

Authors: Jae-Young Ko, Wen-Chin B. Liu, and Patrick P. Mercier

Thanks for your comment. We agree that some readers may be interested in understanding the performance of the circuit as a whole (including active elements) or just the PR device itself. This is why we are now presenting all of the different ways to compute density, so that if readers have specific objectives they are after, they can more easily draw their own conclusions.

To your latter point, if we ignore the VCR of the entire converter and only consider the density of the PR device itself at its $K=1$ operating point, the density we achieve is better than most, but not all, prior work. And we agree this makes sense – this shows that the EFC/AMP techniques we used help improve over a baseline converter with a similarly-performing PR device, but utilizing a better optimized PR device (e.g., operating at higher frequency) can still offer better baseline performance. However, we should note that this is not the point of our work – we are not innovating on a new PR device, but rather, we are proposing a new circuit topology whose purpose is to increase performance of the entire converter at large voltage conversion ratios. Combining the proposed architecture with a better optimized PR device can only further improve the results; we've discussed this possibility in lines 409-416.

3. There are still multiple inaccurate descriptions. (a) The abstract says “baseline PR-based dc-dc topologies have a fixed optimal conversion ratio”, which is a bit misleading. Baseline PR-based topologies achieve flat maximum efficiency during the continuous range of $1/2 < V_{out}/V_{in} < 1$, and this efficiency drops off continuously below $1/2$. This is different than a “fixed” conversion ratio as in switched capacitor converters, so this description should be modified accordingly. (b) Page 2 says “the high coupling and quality factor ($K \times Q$)”, but it's actually $k^2 \times Q$ that is needed for high efficiency as evident in multiple cited works. If the authors are using K to denote a function of k^2 such as $k^2/(1-k^2)$, they should define that. (c) Page 2 says “PRs tend to exhibit optimal performance at the resonant frequency within the inductive region”, but impedance is purely real at the resonant frequency and is only inductive above the resonant frequency. This should instead read “near the resonant frequency”. (d) Perhaps the more relevant PZT comparison to lithium niobate would be using [52] rather than [28].

Thanks for your comment. A) For PR-based DC-DC converters, the definition of optimal VCR indeed differs from that in SC converters. We agree that the description in the abstract could be misleading if interpreted in the context of switched-capacitor converters, so we have

NCOMMS-24-85601-T - Response to Reviewers Comments

Title: A Hybrid Piezoelectric-Resonator-based DC-DC Converter

Authors: Jae-Young Ko, Wen-Chin B. Liu, and Patrick P. Mercier

clarified this point in the abstract. B) We agree and have updated this part accordingly. C) We agree - we have modified this part. D) We agree, and have updated lines 411-414 accordingly.

Reviewer #6

Comments to the Author

All my concerns have been solved, the manuscript can be accepted now.

Thank you for recognizing the value of our approach and findings! We appreciate you taking the time to review the paper carefully and provide constructive feedback.